# Protein structure and folding pathway prediction based on remote homologs recognition using PAthreader

Kailong Zhao[1], Yuhao Xia[1], Fujin Zhang[1], Xiaogen Zhou [1], Stan Z. Li [2✉] & Guijun Zhang [1✉]

Recognition of remote homologous structures is a necessary module in AlphaFold2 and is also essential for the exploration of protein folding pathways. Here, we propose a method, PAthreader, to recognize remote templates and explore folding pathways. Firstly, we design a three-track alignment between predicted distance profiles and structure profiles extracted from PDB and AlphaFold DB, to improve the recognition accuracy of remote templates. Secondly, we improve the performance of AlphaFold2 using the templates identified by PAthreader. Thirdly, we explore protein folding pathways based on our conjecture that dynamic folding information of protein is implicitly contained in its remote homologs. The results show that the average accuracy of PAthreader templates is 11.6% higher than that of HHsearch. In terms of structure modelling, PAthreader outperform AlphaFold2 and ranks first on the CAMEO blind test for the latest three months. Furthermore, we predict protein folding pathways for 37 proteins, in which the results of 7 proteins are almost consistent with those of biological experiments, and the other 30 human proteins have yet to be verified by biological experiments, revealing that folding information can be exploited from remote homologous structures.

[1] College of Information Engineering, Zhejiang University of Technology, HangZhou 310023, China. [2] AI Lab, Research Center for Industries of the Future, Westlake University, Hangzhou 310030 Zhejiang, China. ✉email: Stan.ZQ.Li@westlake.edu.cn; zgj@zjut.edu.cn

AlphaFold2 developed by the DeepMind team has achieved a major breakthrough in machine learning-based protein structure modelling. However, the physics of how proteins dynamically fold into their equilibrium structures is not explored in AlphaFold2[1,2], while understanding protein folding is important for deciphering the genetic code and will promote the exploration of pathogenic mechanism, the development of drug design, and the design of engineered protein-based materials[3–5]. It is well known that templates play a critical role in the protein structure modelling[4]. Meanwhile, the evolutionary relationship implicitly contained in templates are probably favorable for study of protein folding[6].

Templates are essential for improving the accuracy of protein structure prediction. In general, protein structure modelling methods are divided into three categories: physics-based methods, knowledge-based methods, and end-to-end deep learning methods. Physics-based structure modelling methods use the template as the initial structure for folding simulations, such as metropolis monte carlo and molecular dynamics simulation[7]. Knowledge-based modelling approaches usually perform template-based large-scale conformational sampling guided by the energy function, such as RosettaCM[8], D-I-TASSER[9], SWISS-MODEL[10] and MODELLER[11]. In end-to-end deep learning methods, such as AlphaFold2[3], RoseTTAFold[12] and RGN2[13], template information is used explicitly or implicitly for deep learning models. Some recent work have shown that almost all of the popular protein structure prediction methods strongly depend on the quality of the template[2]. For example, it has been reported that the AlphaFold2 structure accuracy in the AlphaFold Protein Structure Database (AlphaFold DB) is largely affected by template availability[4,14]. The distribution of AlphaFold2 models of human proteins with Protein Data Bank (PDB) structures available is heavily skewed to higher average confidence scores than those without PDB structures available[15,16]. There is evidence that the structural space of the PDB is complete and can be used to solve most single-domain proteins[17]. Therefore, developing a new method to recognize high-quality remote homologous templates (PDB structures with a sequence identity <30%)[18,19] is extremely important for protein structure prediction.

Templates are also crucial for the research on protein folding pathways. AlphaFold2 has been successfully used for protein structure prediction using statistical knowledge of the crystal structure. However, it is not clear whether AlphaFold2 can learn the physics of how proteins dynamically fold into equilibrium structure[1,20]. Protein folding is a very fast process (<1 s for small proteins with ≈100 AAs), which results in intermediates that only transiently populate during kinetic folding[21]. Experimentalists usually use spectrometry methods such as circular dichroism chromatography, fluorescence spectroscopy, etc. to study protein folding pathways. Circular dichroism spectroscopy uses the different absorption of left and right circularly polarized light by chiral molecules to analyze protein interactions and determine protein folding paths[22]. Fluorescence spectroscopy is based on the fluorescence emission characteristics of aromatic amino acids in proteins to monitor the folding-unfolding pattern and protein denaturation[23]. These methods can follow kinetic folding but provide very limited information to define the structure of folding intermediates[24]. In some related works, attempts have been made to avoid these difficulties by simulating the folding process. Kresten Lindorff-Larsen et al. studied the folding process of proteins through equilibrium MD simulations by improving the CHARMM force field to make it easier to transfer between different protein classes[21]. Charlotte M Deane et al. use DMPfold to predict the distribution of distances for each residue pair to determine rigid and flexible behavior of proteins[25]. However, inferring the folding path from the mass of simulated data is still a challenge. As a result of gene fusion in the organism, protein folding intermediates may be found in the final state of other proteins[24]. Is it possible to explore folding pathways from a large number of remote homologous structures? For billions of years, organisms in nature have produced stable, functionally safe three-dimensional protein shapes that are reused repeatedly, ranging from short structure to oligomeric complexes[26]. Therefore, all known PDB structures can be grouped into a very limited number of hierarchical families[27,28]. The evolutionary relationships of these protein families may implicitly contain folding information of individual protein. Recently, the Deep-Mind and the EMBL-European Bioinformatics Institute collaborated to create a new data resource, AlphaFold DB, which has attracted the extensive attention of biological scientists[14]. It greatly expands the structural coverage of the known protein sequence space. This makes it possible for AlphaFold DB to complement PDB with respect to the recognition of remote homologous structures and the discovery of template-inspired protein folding pathways. With more than 188,502 publicly available PDB structures (as of August 2020) and 564,449 AlphaFold DB structures (as of March 2022), developing an efficient method is essential for recognizing remote templates and exploring folding pathways.

In the current literature, the common template recognition approaches can be roughly divided into two categories: profile-based alignment methods and binary contact/distance-based threading methods. Most profile-based alignment methods use sequential information, including sequence profile, secondary structure, solvent accessibility and torsion angles to build scoring functions, such as HHsearch[18], SPARKS-X[29] and MUSTER[30]. Binary contact/distance-based threading methods have been developed as great progress has been made by deep learning in inter-residue contacts or distances prediction. For example, in EigenTHREADER[31] and CEthreader[19], eigenvector decomposition is used to resolve the principal eigenvectors, and then an alignment search is performed on the eigenvectors using standard dynamic programming algorithms. DeepThreader[32] aligns the query protein to the template and uses the predicted distance potential to further improve the sequence-template alignment. In addition, there are methods to integrate sequence profile information and contact maps into scoring functions, such as LOMETS[33] and CATHER[34]. These methods improve the accuracy of template recognition to a certain extent, but they can be further improved, as shown in many studies[2,33].

In this work, we proposed a method, PAthreader, to recognize the remote template and explore the folding pathway by learning from PDB and AlphaFold DB. The results on a large-scale benchmark dataset indicate that PAthreader is significantly better than state-of-the-art methods for remote template recognition. To the best of our knowledge, AlphaFold DB is applied to remote template recognition and folding pathway exploration for the first time. We also analyzed the relationship between the template and the model of AlphaFold2, which shows that the quality of model mainly depends on the template availability. When feeding the better template of PAthreader into Alpha-Fold2, the accuracy of its model was improved. Furthermore, we propose a hypothesis that the evolutionary relationship of protein families implicitly contains the folding information of individual proteins. That is, the folding pattern that occur frequently in protein structures are more evolutionarily conserved, and these conserved regions are structurally stable and preferentially formed in protein folding. Based on the above, we identified folding intermediates from homologous templates and obtained protein folding pathways of 7 widely studied cases and 30 human proteins.

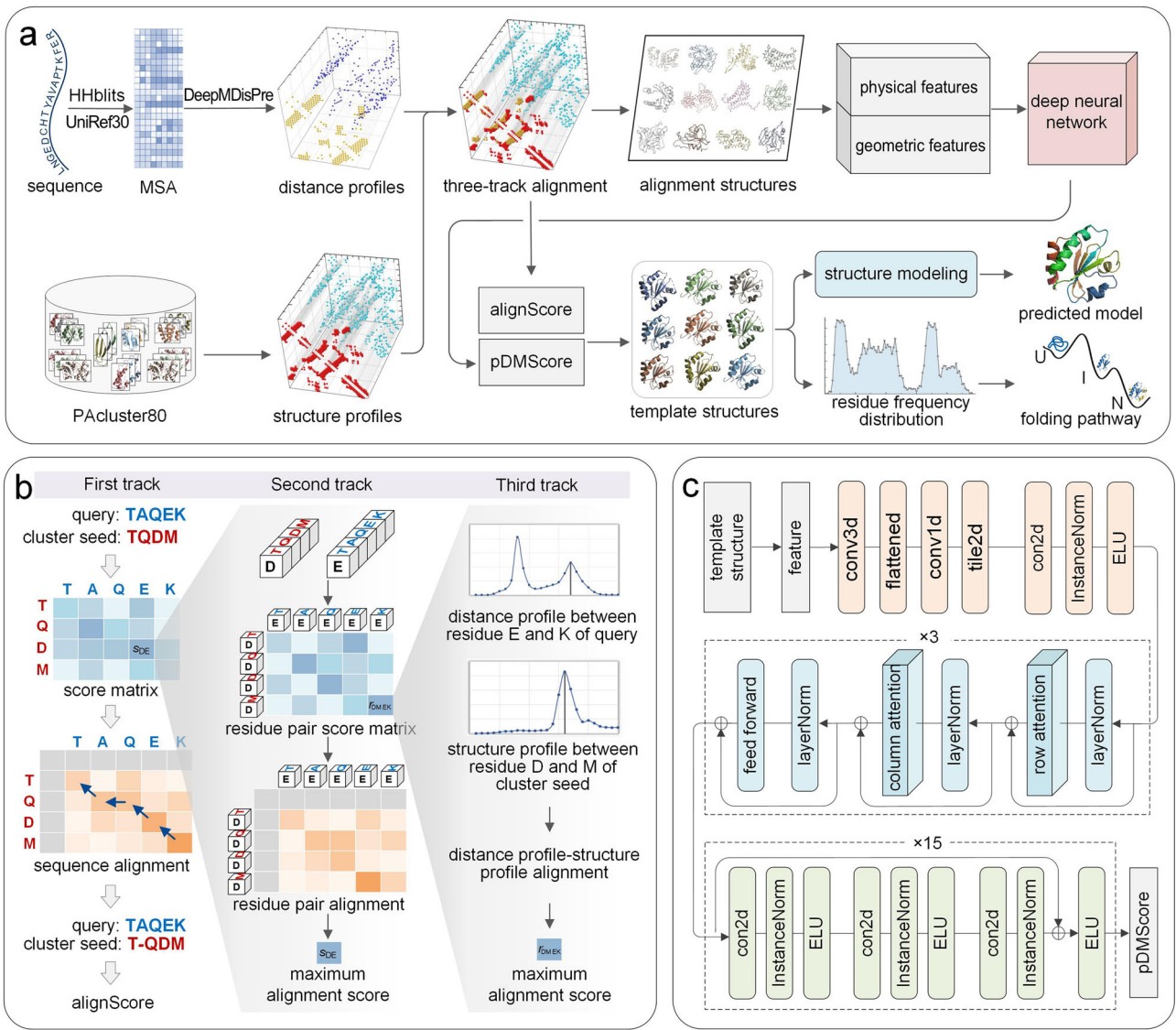

**Fig. 1 Overview of the PAthreader workflow. a** The flowchart of PAthreader. Starting from the sequence, MSA is generated by searching the UniRef30 database using HHblits, and multi-peak distance profiles are predicted by our in-house DeepMDisPre. Meanwhile, structure profiles are extracted from PAcluster80, a master structure database constructed by clustering PDB and AlphaFold DB. Then, a three-track alignment algorithm is designed to align the query sequence to each cluster seed to obtain the maximum alignment score (alignScore). The physical and geometric features of the alignment structures are fed into a trained deep learning model to predict the pDMScore and rank the templates. Finally, the identified templates are integrated into AlphaFold2 for the structure modelling, and the protein folding pathway is determined by identifying folding intermediates according to the residue frequency distribution extracted from templates. **b** Schematic of the three-track alignment. The first track is to calculate the protein-specific score matrix and find the optimal sequence alignment by dynamic programming, where the score matrix is obtained from the second track by finding the optimal residue pair alignment. Residue pair alignment is performed based on the construction of the residue pair score matrix, where the values are calculated from the third track by maximizing the product of probabilities and minimizing the distance difference. **c** The deep neural network for pDMScore prediction, which consists of 3 axial attention blocks and 15 residual blocks.

## Results

**PAthreader overview**. The pipeline of PAthreader is illustrated in Fig. 1, and the details are presented in the Methods section. First, multi-peak distance profiles are obtained by our in-house DeepMDisPre, which may predict multiple possible distances for flexible protein regions. Structure profiles are extracted from PAcluster80, a master structure database built by clustering PDB and AlphaFold DB with a threshold of 80% structural similarity. Then, a three-track alignment algorithm is proposed to align the query sequence to each cluster seed of PAcluster80, in which the protein-specific score matrix is first calculated by residue pair alignment and profile alignment, and the optimal sequence alignment is then searched by dynamic programming and the

maximum alignment score (alignScore) is obtained. Subsequently, physical and geometric features are extracted from the alignment structure and fed into a convolutional network with self-attention to predict the DMScore (pDMScore)[35], a global structure scoring metric that is complementary to alignScore and linearly weighted with the alignScore for template ranking. Finally, the top templates are fed into state-of-the-art protein structure prediction methods for structure modelling. Meanwhile, the folding pathway is explored based on the folding intermediates, which are deduced according to secondary structures and frequency distributions of residues calculated on the basis of different distance deviation thresholds after template alignment by TM-align[36].

**Table 1 TM-score of template recognition on 551 tested proteins.**

|            | (0.9, 1.0) | (0.7, 0.9) | (0.5, 0.7) | (0.0, 0.5) | All       |
|------------|------------|------------|------------|------------|-----------|
|            | num (58)   | num (321)  | num (149)  | num (23)   | num (551) |
| PAthreader | **0.899**  | **0.787**  | **0.568**  | **0.424**  | **0.725** |
| HHsearch   | 0.840      | 0.718      | 0.476      | 0.272      | 0.646     |
| LOMETS3    | 0.868      | 0.754      | 0.534      | 0.342      | 0.689     |

Tested proteins were divided into four subsets (0–0.5, 0.5–0.7, 0.7–0.9 and 0.9–1) based on TM-score of the best template of targets in PDB. Bold text highlights the best result in each category.

**Fig. 2 Performance of PAthreader for template recognition. a, b** Head-to-head TM-score comparison of PAthreader with HHsearch and LOMETS3 at different difficulty levels. **c** Average TM-score on single-domain and 2-domain and ≥3-domain proteins, with corresponding protein numbers in parentheses.

**PAthreader significantly outperforms state-of-the-art methods for template recognition.** We constructed a test set of 551 nonredundant proteins with sequence identities <30% from the SCOPe[37] database. The resolutions of these proteins are less than 2 Å, and their lengths range from 120 to 700 residues. We compared our method with state-of-the-art threading protocols, including HHsearch[18] and LOMETS3[33], where homologous templates with a sequence identity ≥30% to the query were excluded. In order to objectively evaluate the performance of these methods at different difficulty levels, we divided the test set into 4 subsets based on the TM-score of the best template in PDB, i.e., 0–0.5, 0.5–0.7, 0.7–0.9 and 0.9–1 (Supplementary Data 1). The results are summarized in Table 1. Our method outperforms HHsearch and LOMETS3, with the average TM-score of test targets that is 12.2% and 5.2% higher than that of HHsearch and LOMETS3, respectively. The $P$ values of Student's $t$-test are 3.29E-47 and 1.52E-25, respectively, indicating statistically significant differences between them. Figure 2a and b present the head-to-head TM-score comparison between methods, where PAthreader achieves a higher TM-score than HHsearch and LOMETS3 for 76% and 71.3% of the targets, respectively. In particular, the performance of PAthreader is significantly better than that of HHsearch and LOMETS3 for the hard targets (0–0.5 and 0.5–0.7). We compared HHsearch/LOMETS3 and PAthreader without AlphaFold DB, and the results are shown in Supplementary Table S1. The average TM-score of the first template identified by PAthreader is 0.702, which is 8.6% and 1.9% higher than that of HHsearch and LOMETS3, respectively. We also compared our method with SPARKS-X、MUSTER、CEthreader and EigenTHREADER, where homologous templates with a sequence identity ≥30% to the query were excluded. The results are summarized in Supplementary Table S2. The average TM-score of PAthreader is 12.2%, 5.2%, 8.4%, 8.5%, 7.9% and 10.5% higher than that of HHsearch、LOMETS3、SPARKS-X、MUSTER、CEthreader and EigenTHREADER, respectively. The results show that the PAthreader performs better than these methods.

Until now, achieving accurate predictions for multi-domain protein structures has been more challenging than for single-domain proteins[38,39]. Therefore, we compared PAthreader with HHsearch and LOMETS3 in terms of template recognition ability for multi-domain proteins. Here, the test set was classified into three categories: single-domain proteins (1dom), proteins with 2 domains (2dom), and proteins consisting of ≥3 domains (≥3dom). The results are reported in Fig. 2c. From this figure, it is shown that the average TM-score of PAthreader is 12.2% and 4.1% higher than that of HHsearch and LOMETS3 for single-domain proteins, respectively. For multi-domain proteins, the average TM-score of PAthreader is 12.3% and 7.9% better than that of HHsearch and LOMETS3, respectively. These results demonstrate the robustness of PAthreader, which outperforms state-of-the-art template recognition methods for both single-domain and multi-domain proteins. Moreover, we attempted to recognize the templates of protein complexes by adding 21 residue repeated Glycine-Glycine-Serine to link the chains of the complexes together[40]. Supplementary Fig. S1 shows that templates with TM-scores = 0.97 and 0.91 are obtained for 1AB9 and 1GRN, respectively, using PAthreader. These results suggest that although PAthreader was designed for single-chain proteins, it may be used to recognize templates of protein complexes.

There may be three reasons for the better performance of PAthreader over HHsearch and LOMETS3. Firstly, AlphaFold DB helps to improve the accuracy of template recognition by expanding the family coverage of model organism proteomes. From the structures of 48 species provided by AlphaFold DB, which provides novel folding architectures and motifs not found in PDB, we selected 100,912 high-confidence structures[15] with a predicted local distance difference test (pLDDT) score ≥90 to complement the master structure database from PDB. We investigate the effect of AlphaFold DB on template recognition through ablation experiments, and the results are shown in Fig. 3a and Supplementary Table S1. When AlphaFold DB is used, the average TM-score of PAthreader is 0.1%, 1%, 8.2% and 23.4% higher than those without using AlphaFold DB for the targets of 0.9–1, 0.7–0.9, 0.5–0.7 and 0–0.5, respectively, where the performance of PAthreader is significantly improved for the hard targets (0–0.5 and 0.5–0.7). However, the performance of

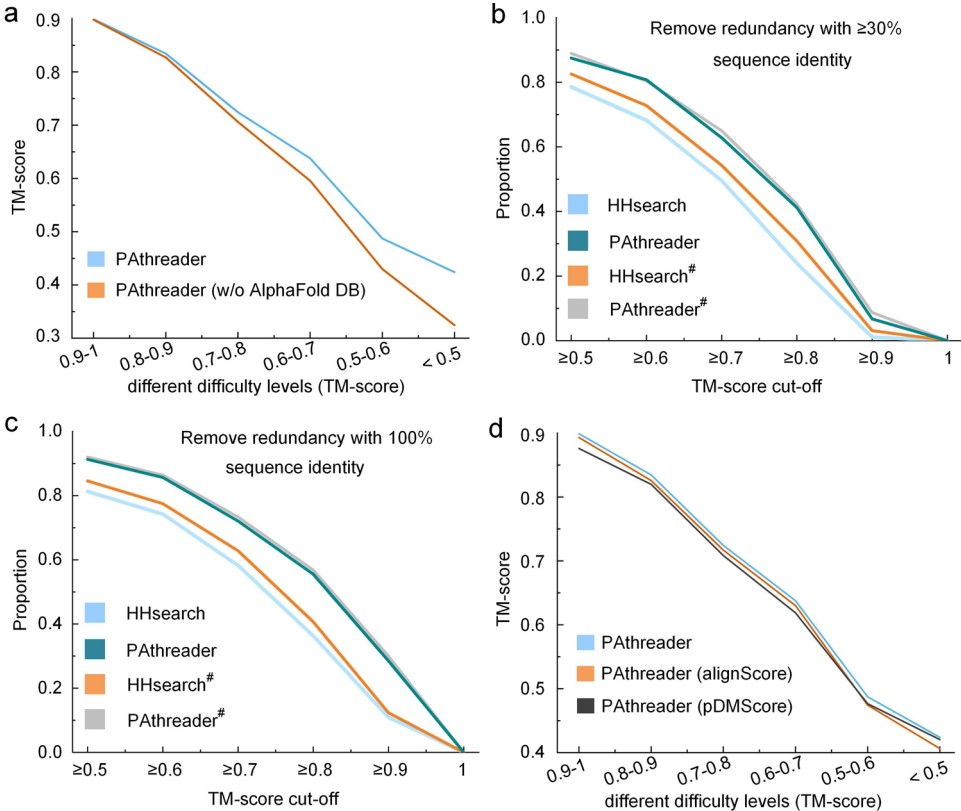

**Fig. 3 Performance of PAthreader for template recognition. a** The average TM-score for template recognition with and without AlphaFold DB at different cut-off ranges. **b, c** The proportion of the number of templates with ≥30% and 100% sequence identity removed at different TM-score cut-off. PAthreader[#] and HHsearch[#] represent the results obtained by comparing the identified structure with the native structure through TM-align. **d** The average TM-score for different template rankings, showing the effect of pDMScore on template recognition.

PAthreader is not remarkably improved for the easy targets (0.7–0.9 and 0.9–1) because they already have promising templates in the PDB. This suggests that AlphaFold DB may be an effective complement to the current PDB, especially for extremely remote templates, which is helpful to improve the accuracy of template recognition. Supplementary Fig. S2a shows an example, 1HZ4_A, whose TM-score is significantly improved from 0.56 to 0.84 due to the extension of AlphaFold DB.

Secondly, the three-track alignment takes full advantage of the structural information and predicted multiple promising distances for residue alignment, residue pair alignment, and distance-structure profile alignment. In order to analyze the performance of the three-track alignment, we perform sequence-structure threading for the identified structures in different ways on the benchmark set. Figure 3b presents a comparison, where the results of PAthreader are obtained by threading the sequences into the identified structures based on the sequence alignment provided by the three-track alignment, and the results of HHsearch are generated by the Hidden Markov-constructed profiles comparison. PAthreader[#] and HHsearch[#] represent the results obtained by comparing the identified structure with the native structure through TM-align, which is denoted as the native alignment. The comparison of PAthreader[#] and HHsearch[#] shows that the template structures identified by our method are better than those of HHsearch. It also highlights that there is a gap between HHsearch and its native alignment HHsearch[#], while PAthreader is almost close to its native alignment PAthreader[#]. These results indicate that more accurate structures are identified when using our method, and more accurate sequence-template alignments than HHsearch are presented. The performance is further confirmed by the comparison of the

results shown in Fig. 3c, where PAthreader also significantly outperforms HHsearch when removing homologous templates with 100% sequence identity. Supplementary Fig. S2b shows a representative example, where 3CES_A is identified as the best template structure for the query target using both PAthreader and HHsearch. However, the alignment provided by PAthreader is closer to the native alignment than HHsearch, resulting in a better TM-score for the PAthreader template (0.81) than the HHsearch template (0.55).

Thirdly, we combine the pDMScore predicted by deep learning and the alignScore obtained from the three-track alignment to select better templates. Figure 3d and Supplementary Table S1 present the results of the ablation experiments. Compared with alignScore, the templates that are selected by pDMScore are better for cases at the difficulty level of 0-0.5. This is because pDMScore uses a deep neural network that combines physical and geometric features of structures, reducing the noise from AlphaFold DB and the predicted distance profiles and effectively complementing to the alignScore. Therefore, when using the score based on the linear weighting of alignScore and pDMScore, the template is better than that selected by independently using alignScore and pDMScore. As shown in Supplementary Fig. S2c, the TM-score of the template identified by the combination of alignScore and pDMScore is 0.89, which is higher than that identified by alignScore (0.66) and pDMScore (0.64) obtained independently, since the template has accurate domain orientation and high coverage. Furthermore, the average run time of PAthreader is 1.25 hours for 551 test proteins when computing on a single workstation with 10 Intel Xeon Silver 4210 R 2.4 GHz CPUs. Supplementary Fig. S3 shows the runtime of each protein, which increases nearly linearly with sequence length.

**Our results suggest that the performance of AlphaFold2 depends on the quality of the template**. AlphaFold2 has made a great advance in protein structure prediction based on the current sequence library and structure library. Here, we performed two experiments to analyze the relationship between the template and the accuracy of the AlphaFold2 model. Firstly, we examined the quality of templates for high-scoring models (pLDDT ≥ 90) of AlphaFold DB by structural comparison. As of March 2022, AlphaFold DB provides 564,449 structures for 48 organisms. We clustered 100,912 high-scoring models selected from AlphaFold DB using TM-align with 80% structural similarity. We found that 55.7% of the structures could be classified into the 34,701 PDB clusters determined during the construction of the master structure database, and the remaining structures could form 22,105 new clusters. This indicates that more than half of the high-scoring models in AlphaFold DB have templates with TM-score ≥ 0.8 in PDB.

Secondly, we further analyzed the relationship between the model accuracy and the first template used for AlphaFold2 modelling on 551 test proteins, where all templates with sequence identity ≥30% were removed. The results are presented in Supplementary Fig. S4a. On the 274 targets with higher quality templates (TM-score ≥ 0.7), models with TM-score ≥ 0.9 are generated for 88% of targets by AlphaFold2. On 277 targets with relatively poor templates (TM-score < 0.7), the number of models with TM-score ≥ 0.9 predicted by AlphaFold2 decreased to 49.8%. However, in some cases without high quality templates, high scoring models can still be obtained by AlphaFold2. We speculate that there may be two main reasons for this. One is that these targets can be searched for abundant multiple sequence alignments (MSAs), which are used to infer the precise residue relationship by AlphaFold2. The other is that these test proteins may be included in the training set of AlphaFold2, which makes it difficult to accurately test the effect of templates on AlphaFold2[4]. We also used the CAMEO test set (2022/04/01-2022/06/18) to analyze the relationship between the template and the accuracy of the AlphaFold2 model. The results are presented in Supplementary Fig. S4b. The CAMEO test set has the same trend as the 551 test proteins in terms of the relationship between the template and the accuracy of the AlphaFold2 model. AlphaFold2 produced more and more low-scoring models as template quality decreased.

The above results suggest that the performance of AlphaFold2 depends on the quality of the template to some extent. Therefore, it is possible to improve AlphaFold2 by providing better templates.

**AlphaFold2 model can be enhanced by PAthreader templates**. We participated in the CAMEO blind test by replacing the template recognition component (HHsearch) of AlphaFold2 with PAthreader. In a continuous three-month test, our method (PAthreader) achieved better results than the state-of-the-art methods, and ranked first among the public servers (Supplementary Table S3). Here, we compare our method with Alpha-Fold2 and RoseTTAFold on the CAMEO targets. The results of AlphaFold2 were obtained by running the standalone package locally and those of PureAF2_orig, PureAF2_notemp, and RoseTTAFold were obtained from the CAMEO official website (https://www.cameo3d.org/modelling/). Figure 4a and Table 2 present the results, where PAthreader obtains the better model than other methods on most of the cases and achieves higher TM-score compared to other methods on average. This suggests PAthreader was able to provide better templates for the modelling than HHsearch. Figure 4b presents the template recognition results of PAthreader and HHsearch, where the average TM-score of PAthreader is 11.2% higher than that of HHsearch on the

CAMEO targets. PAthreader generates templates with TM-score ≥ 0.9 on 32.3% of the targets, which is almost twice as high as HHsearch (17.2%). We analyzed the relationship between the model accuracy and the first template used for PAthreader modelling on CAMEO targets (Supplementary Data 2). On the 62 targets with poor quality templates (TM-score < 0.7), models with TM-score ≥ 0.9 are generated for 38.7% of targets by PAthreader. On 124 targets with higher templates (TM-score ≥ 0.7), the number of models with TM-score ≥ 0.9 predicted by PAthreader increased to 83.1%. Interestingly, we find that PAthreader obtains templates with a higher TM-score than AlphaFold2 models on 10.2% targets (Fig. 4c). These results again demonstrate the ability of PAthreader in terms of template recognition.

The number of structural patterns or family types of proteins is limited. Many proteins with low sequence identity correspond to the same structural pattern in PAcluster80. Therefore, templates are essential for AlphaFold2 modelling when a protein does not have enough MSAs to infer atomic coordinates and its corresponding structural pattern happens to exist. Figure 4d presents two representative examples, a single-domain protein (7PNO_D) and a multi-domain protein (7T4Z_A). For the target 7PNO_D, PAthreader generates a model with TM-score of 0.89, which is better than that of AlphaFold2 (0.75) and PureAF2_orig (0.73). This is because there is only one sequence in the MSA, which cannot probably provide sufficient co-evolutionary information to infer residues relationship, resulting in the protein modelling of AlphaFold2 being heavily dependent on the template. However, HHsearch employed by AlphaFold2 provided a poor template with TM-score = 0.33. PAthreader identified a better template with TM-score = 0.64, which results in a better final model. For target 7T4Z_A, PAthreader generates a model with TM-score of 0.92, which is better than that of AlphaFold2 (0.84) and PureAF2_orig (0.84). This is because PAthreader recognize a better template with TM-score = 0.85 than that of HHsearch (0.77), which provide accurate domain orientations for multi-domain proteins modelling.

Since 2019, considerable efforts have been made to determine the structure of proteins in SARS-CoV-2, a novel coronavirus responsible for the COVID-19 pandemic. We used PAthreader to recognize templates and model structures for 17 proteins of SARS-CoV-2 virus, and the results are shown in Supplementary Table S4. The average TM-score of PAthreader templates is 0.725, which is 11.3% higher than that of HHsearch. In structural modeling, PAthreader models achieved a slightly higher average TM-score (0.825) than the AlphaFold2 models (0.812). In Supplementary Fig. S5, we show a comparison of structural model built by PAthreader and AlphaFold2, where the PAthreader model has the TM-score of 0.987 which is better than that of the AlphaFold2 model (TM-score = 0.825).

**Protein folding pathway exploration**. PAthreader predicts the protein folding pathways based on intermediates inferred from remote homologous proteins, which likely originate from a common ancestor and can reveal the evolutionary relationship between proteins[27,41]. It is well known that the structures of proteins are more evolutionarily conserved than their sequences, and amino acid mutations may not result in major changes in structure and function[42,43]. When comparing remote homologous proteins with each other, it can be found that there are highly similar structures between them that are very stable in organisms after billions of years of evolution. Inspired by these ideas, we speculate that folding intermediates may be related to these stable structures and that they may have been selected by evolution so as to have a rapid and robust pathway for folding to the native structure. In other words, the evolutionary

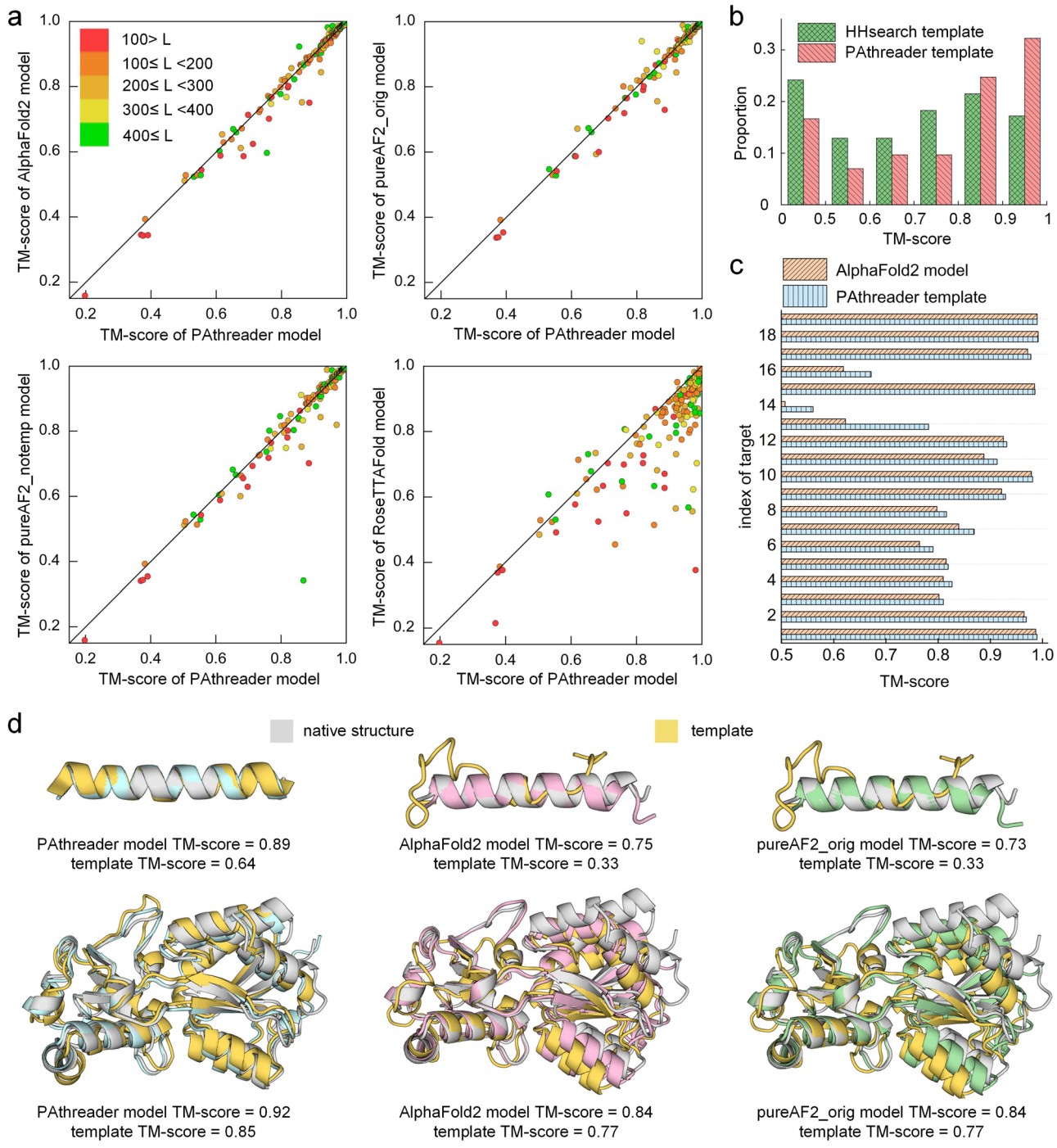

**Fig. 4 Performance of PAthreader on CAMEO. a** Head-to-head TM-score comparison of PAthreader with AlphaFold2, pureAF2_orig, pureAF2_notemp, and RoseTTAFold for structure modelling. Each point represents a protein target, and different colors indicate different protein sizes. **b** The distribution of the TM-score of templates identified by PAthreader and HHsearch. **c** Comparison of the TM-score of templates by PAthreader and the model obtained using AlphaFold2 on 19 proteins. **d** Examples of the single-domain protein 7PNO_D and the multi-domain protein 7T4Z_A. The structure superpositions of PAthreader model (blue), AlphaFold2 model (pink), and pureAF2_orig model (green) with the native structure (grey) and template (yellow) are shown.

relationships of protein families may contain folding information of individual protein, such as folding intermediates. In addition, there are some reports that the secondary structure influences folding and unfolding kinetics and could determine the folding pathway[23], especially the pattern imposed by the cooperative formation of α-helices and β-sheets, which may be an important factor in determining the folding pathway[44,45]. Based on the above assumptions, we explored protein folding pathways from

homologous templates and secondary structures, consisting of two consecutive steps. First, the global structural alignment of the identified homologous templates with the first template was performed by TM-align. The residue frequency scores (ResFscore) were then calculated at different distance deviation thresholds, where a higher value indicates a higher frequency of residue alignment at corresponding positions of template structures. Second, the target was divided into multiple segments based

on the secondary structure, and the average ResFscore for each segment was calculated by dividing the corresponding area of the segment by its length. Folding intermediates were finally determined by adaptively selecting fragments with an average ResFscore greater than a given threshold (see details in Methods section). We explored protein folding pathways on 7 widely studied cases and 30 human proteins (Supplementary Data 3). The results show that the 7 proteins are almost consistent with

biological experiments, and the other 30 human proteins have yet to be verified by biological experiments.

Horse heart cytochrome c is a member of the cytochrome c superfamily. It is an electron-transfer protein with a heme c group, and it binds to protein via thioether bonds to complete the transfer of electrons. The structure consists of five α-helices connected by random coil segments, which fold around the heme moiety with histidine residue (H18) and methionine residue (M80) as axial ligands[46]. In the current literature, two different folding pathways of this protein have been identified by hydrogen exchange (HX) pulse labeling and nuclear magnetic resonance (NMR). Figure 5a shows the first experimental pathway, where the blue helices are first folded and followed by red regions[45]. Figure 5b shows the second experimental pathway, where blue is folded first, followed by green, yellow, red and then grey[47,48]. It contains 4 folding intermediates, $I_1$, $I_2$, $I_3$ and $I_4$ (Supplementary Fig. S6).

Figure 5c shows the folding intermediate and pathway of horse heart cytochrome c predicted by PAthreader, where the intermediate contains three helices that are consistent with the first experimental pathway. Although multiple intermediates of the second experimental pathway are not accurately predicted, the ResFscore distribution of residues identified by PAthreader from homologous templates can explain the folding order determined by biological experiments well. As shown in Fig. 5d, the N- and C-terminal helices corresponding to blue segments of

**Table 2 TM-score of structure modelling and template recognition on the three-month CAMEO blind test.**

|  | 2022/04/01–2022/04/23 | 2022/04/29–2022/05/21 | 2022/05/27–2022/06/18 |
|---|---|---|---|
| Structure modelling |  |  |  |
| PAthreader | **0.882** | **0.900** | **0.876** |
| AlphaFold2 | 0.878 | 0.896 | 0.869 |
| pureAF2_orig | 0.877 | 0.892 | \ |
| pureAF2_notemp | 0.864 | 0.896 | \ |
| RoseTTAFold | 0.811 | 0.824 | 0.830 |
| Template recognition |  |  |  |
| PAthreader | **0.719** | **0.754** | **0.776** |
| HHsearch | 0.650 | 0.672 | 0.690 |

The results of AlphaFold2 were obtained by running the standalone package locally and those of PureAF2_orig, PureAF2_notemp, and RoseTTAFold were obtained from the CAMEO official website. Bold text highlights the best result in each category.

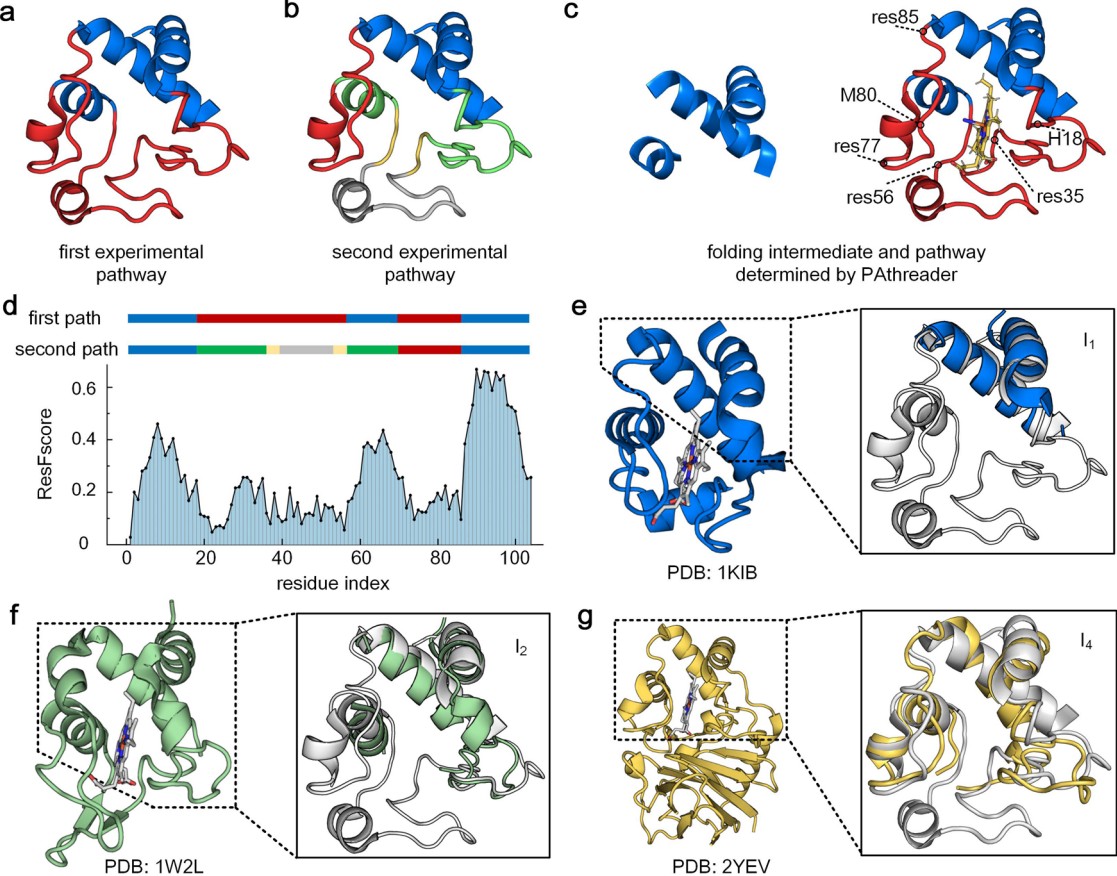

**Fig. 5 Folding pathway of horse heart cytochrome c (PDB ID: 1I5T). a** The first experimental pathway. The blue region is first folded and is followed by the red region. **b** The second experimental pathway. Blue is folded first, followed by green, yellow, red and then grey. It contains 4 intermediates, $I_1$ (blue), $I_2$ (blue + green), $I_3$ (blue + green + yellow) and $I_4$ (blue + green + yellow + red). **c** Intermediate and folding pathways predicted by PAthreader, the blue region is first folded and is followed by the red region. **d** Two different experimental paths and the ResFscore distribution of residues identified by PAthreader. **e–g** Template structures from 1KIB, 1W2L and 2YEV. The solid line box is the partial superposition of templates and the structure of horse heart cytochrome c (grey), which correspond to intermediates of the second experimental pathway.

the second experimental pathway have a relatively high ResFscore, followed by the green segments. The red and grey segments have the lowest ResFscore.

The order of the yellow segments could not be identified accurately because they are too short. This result indicates that the average ResFscore corresponding to the secondary structure can reveal the folding order of the protein. From the analysis of the results, we found that intermediates could be found in the template structures identified by PAthreader. Figure 5e–g show similar template structures of intermediates $I_1$, $I_2$ and $I_4$ of the second experimental pathway (the difference between $I_3$ and $I_2$ is not obvious), which are from the cyanobacterium Arthrospira maxima cytochrome c6 (PDB ID: 1KIB), Rhodothermus marinus caa3 cytochrome c domain (PDB ID: 1W2L) and Thermus thermophilus caa3-type cytochrome oxidase (PDB ID: 2YEV) in PDB, respectively. It can be seen that the three structures correspond to different intermediates of the second experimental pathway. They are from the same protein family and have similar electron transfer functions. However, these structures have been changed during the evolution of different organisms, which are mainly reflected in local structures other than preferentially folded three helices. For example, compared with the bottom loop of cytochrome c (residues 35–56 in Fig. 5c), 1KIB and 1W2L form a short helix and two antiparallel β-strands in the corresponding regions, respectively. Based on the differences between these structures, we further analyzed why the three helical segments of cytochrome c are preferentially folded. The local structural changes of the cytochrome c family are probably related to the electron transfer function, since ligand binding can induce conformational changes[49], which are characterized by heme crevice undergoing closed-open transitions concomitant with shifts of residues 77–85 and residues 35–56[46]. These flexible local structures have a high probability of mutation in biological evolution, while the pocket region formed by the three helices is stable. Therefore, the order of protein folding is related to its function, and relatively stable regions of structure are preferentially formed during protein folding. These results suggest that the final state structures of protein families may be the folding intermediates of individual protein, which implicitly contain folding information. Interestingly, we also found an intermediate-like structure of cytochrome c in the complex, which is composed of the N-terminal of chain 1 and the C-terminal of chain 2 (Supplementary Fig. S7). This suggests that proteins in organisms, including single-domain proteins, multi-domain proteins and complexes, may be formed with a limited number of folding patterns.

Figure 6 shows the results of the folding pathways of synaptic protein PSD-95 (PDB ID: 1BE9), T4 lysozyme (PDB ID: 2LZM), human protein ckshs1 (PDB ID: 1DKT), apomyoglobin (PDB ID: 1MBC), acyl-CoA binding protein (PDB ID: 1NTI) and redesigned Rd-apocyt b562 based on apocytochrome b562 (PDB ID: 1YYJ)[50–55]. Figure 6a and b show the folding pathway determined by the biological experiments, where the blue region is folded first and is followed by the red region (the yellow region is not clearly defined). Figure 6c and d show the ResFscore distribution and folding pathway identified by PAthreader. The results show that the ResFscore distribution identified by our method is consistent with the experiment determined protein folding order for all proteins. It can be seen that relatively high ResFscore often correspond to segments of α-helices or β-sheets, while ResFscore of the loop regions that connect α-helices or β-sheets have low ResFscore. This is consistent with the conclusion that has been reported in the literature, that is, the folding nucleus formed by the secondary structure is important in determining the folding pathway of the protein[44,45], and the folding

intermediate contains extensive secondary structures. In addition, the secondary structure segments contained in the intermediates identified by PAthreader are consistent with the results of biological experiments for the first five proteins. For the last protein, 1YYJ, the intermediate identified by PAthreader contains three helices, which is slightly different from that determined by native-state hydrogen exchange. This may be because 1YYJ is an artificially designed protein, which results in its evolutionary information not being accurately obtained from the protein family. Figure 6e shows the template structures identified by PAthreader, which have a similar intermediate structure to the target proteins. The results of the 30 human proteins studied are summarized in Fig. 7. Supplementary Fig. S8 and Supplementary Table S5 show the optimal templates, folding intermediates, and the proportion of templates used to recognize the intermediates in different pfam families in detail. These results reveal that the evolutionary relationships of protein families contain folding information of the individual protein, which may provide a basis for understanding the function and mechanism of molecules and guide experimental structure determination.

## Discussion

In this study, we developed PAthreader for remote template recognition and template-based protein structure prediction and folding pathway exploration. The results show that PAthreader outperforms the state-of-the-art methods HHsearch and LOMETS3 for remote template recognition. The ability of PAthreader to recognize better remote templates is mainly attributed to three aspects: the family coverage of model organism proteomes is expanded through AlphaFold DB, the three-track alignment algorithm provides accurate sequence-template alignment, and the predicted pDMScore helps to select physically plausible templates. PAthreader can complete template recognition for proteins <120 AAs in 0.5 h using 10 CPUs, which is acceptable for most applications. This is mainly attributed to the clustering of the master structure database, which provides structure profiles for alignment to help quickly locate the best structural classes.

We demonstrated that the state-of-the-art protein modelling method AlphaFold2 depends on the quality of templates to some extent. Especially for multi-domain proteins, the model predicted by AlphaFold2 still has a large deviation due to inaccurate domain orientations[56]. We further enhanced AlphaFold2 using PAthreader and ranked it first in the CAMEO blind test for three consecutive months (2022/04/01-2022/06/18). However, we found that directly providing better templates can only improve AlphaFold2 to a certain extent, which may be the reason for the over-engineering of AlphaFold2. Embedding PAthreader into AlphaFold2 and retraining it may obtain a more suitable model, but it would be a challenge because it requires an extremely large number of computational resources.

Based on the recognized homologous templates, we further explored protein folding pathways by identifying folding intermediates. In this study, we proposed that the evolutionary relationships of protein families implicitly contain folding information of individual protein, and protein folding intermediates can be inferred from collections of remote homologous structures. We explored the protein folding pathways of 37 proteins. The results of 7 proteins are almost consistent with biological experiments, and the remaining 30 human proteins have yet to be verified by biological experiments. The ability of PAthreader to explore folding pathways is mainly attributed to two aspects. First, AlphaFold DB complements a large number of remote templates, which provide sufficient evolutionary information

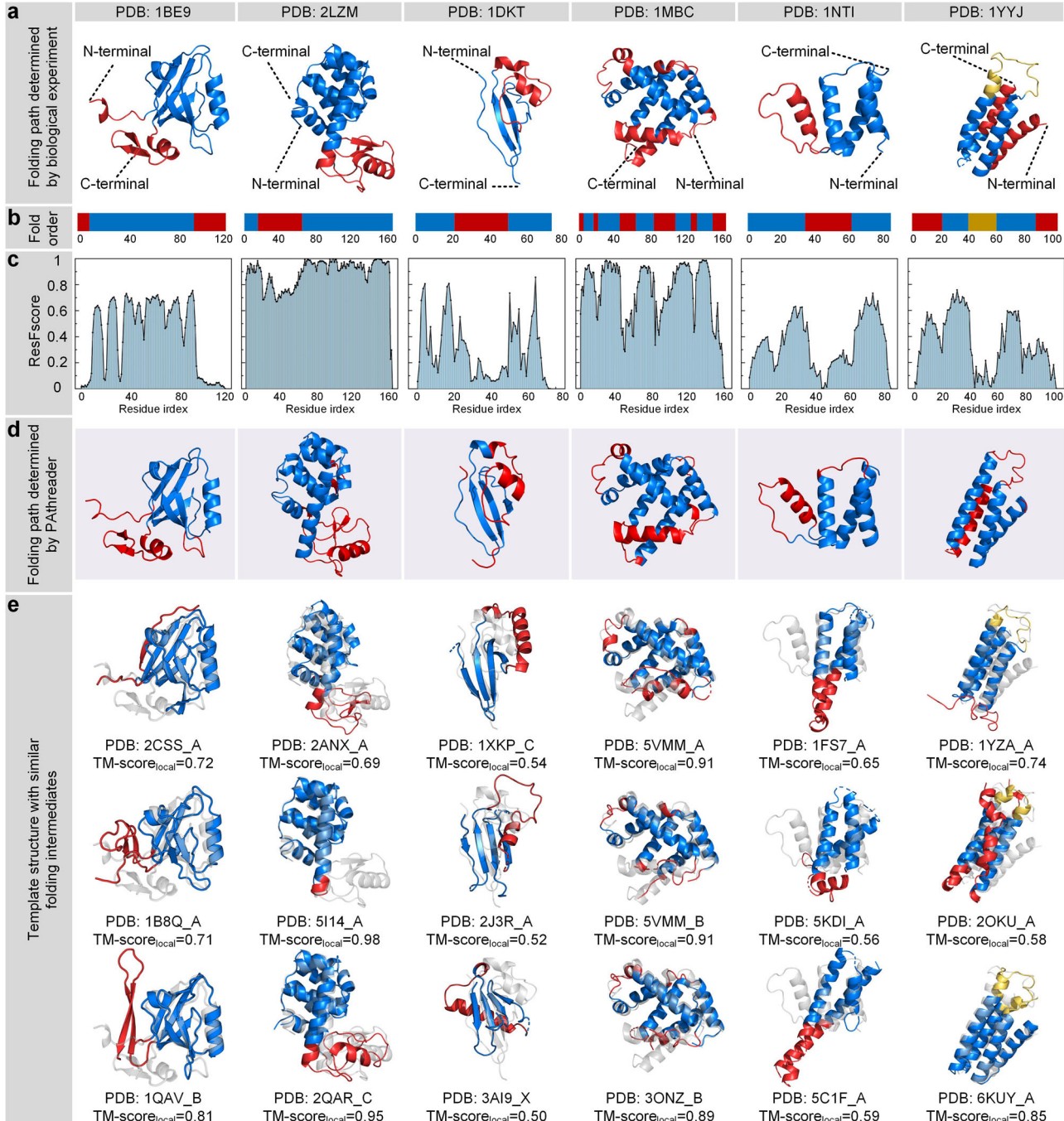

**Fig. 6 Results of protein folding pathways. a**, **b** Folding pathway determined by biological experiments. The folding order is blue and then red. **c** The residue frequency distribution identified by PAthreader. **d** Folding pathway determined by PAthreader. **e** Template structures with folding intermediates (blue) that are similar to those of the target protein (grey). TM-score$_{local}$ is the similarity between the local structure (blue) of the template and the target protein.

between protein families. The second important contribution to PAthreader is that DeepMDisPre predicts multiple promising distances for flexible protein regions (an example is shown in Supplementary Fig. S9), and the three-track alignment uses predicted multi-peak distance profiles to optimize the alignment, which allows templates that satisfy all possible distance constraints to be recognized. The exploration of folding pathways based on remote templates provides new ideas and insights for the study of protein folding mechanisms, which will be advanced by the accumulation of structures deposited in PDB and the rapid expansion of AlphaFold DB.

## Methods

**Benchmark set**. The benchmark set was constructed based on SCOPe 2.07[37], which was divided into 11,198 clusters by CD-HIT[57] with a 30% sequence identity cut-off. We selected 2021 clusters with only one member as candidate sets because they have few templates with high sequence identity in the template library, which helps to objectively evaluate the ability of remote templates recognition. In the 2021 clusters, 551 nonredundant proteins were selected as the benchmark set based on sequence lengths ranging from 120 to 700 AAs and resolutions less than or equal to 2.0 Å. The parameters of PAthreader are listed in Supplementary Table S6.

**Master structure database**. We constructed a master structure database, named PAcluster80, based on PDB and AlphaFold DB, and the flowchart is shown in

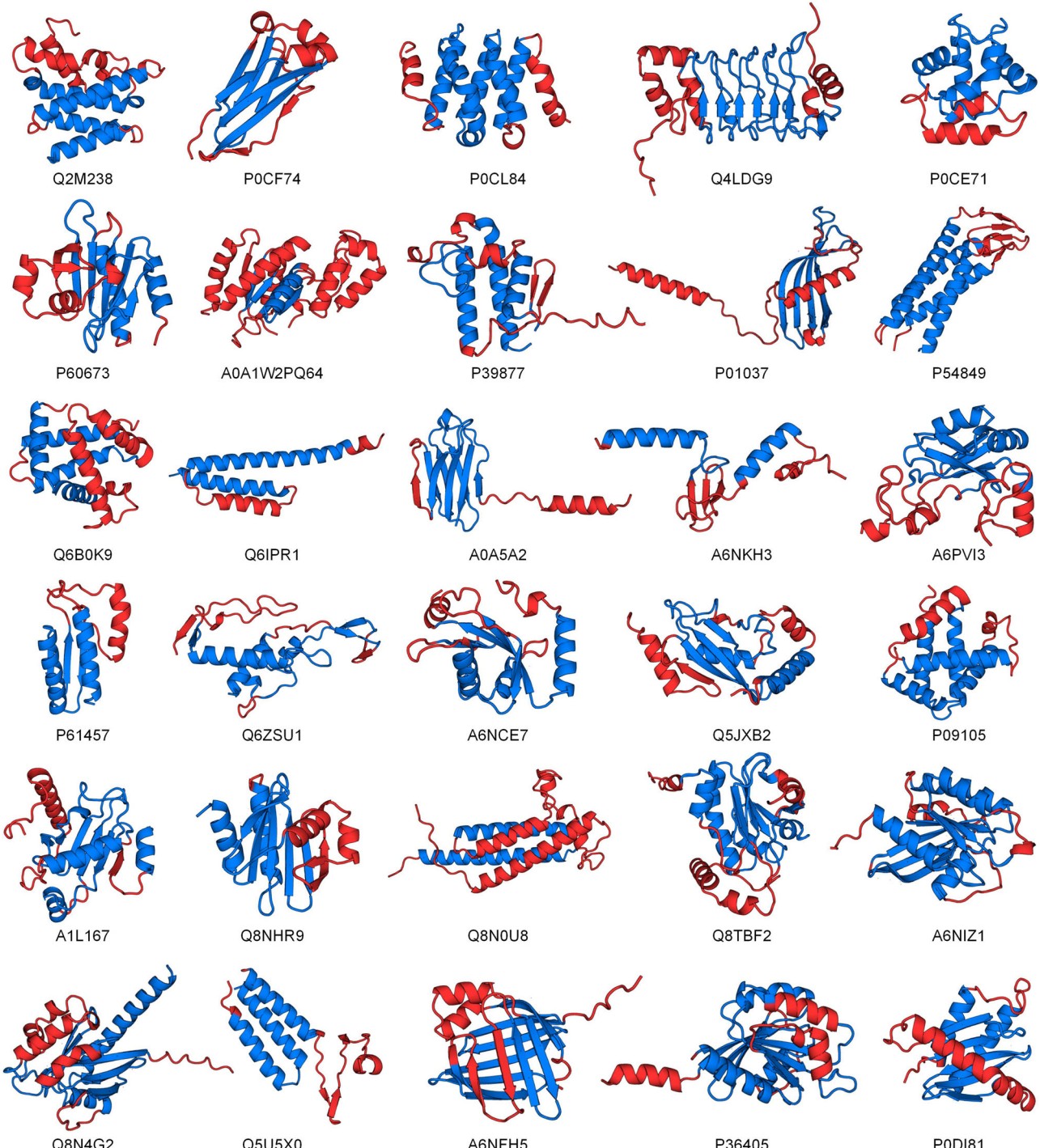

**Fig. 7 Results of folding pathways predicted by PAthreader on 30 human proteins.** Thirty human proteins, whose native structures have not been determined by biological experiments are labeled with their UniProt accession. The structures shown are identified by template recognition. The blue region is the intermediate, and the folding order is blue and then red.

Supplementary Fig. S10. Firstly, we removed structures with 100% sequence identity from the PDB, since identical sequences often correspond to very similar structures. Then, we calculated the structural similarity of the retained 106,275 proteins using TM-align and classified them into 34,701 structural classes based on an 80% structural similarity threshold (protein pairs with TM-score ≥ 0.8 have a great probability of being the same SCOP fold, and they have very similar topologies)[58]. In this process, we used a greedy incremental clustering approach similar to CD-HIT, which avoids many pairwise structure alignments (see details in Supplementary Note S1). As of March 2022, there were 564,449 predicted structures from 48 species provided by AlphaFold DB (https://alphafold.ebi.ac.uk/). We selected 100,912 structures with pLDDT ≥ 90 from these structures as available templates and clustered them using the 34,701 PDB cluster seeds according to 80%

structural similarity, resulting in 55.7% of the structures being classified into 34,701 PDB clusters and the remaining structures forming 22,105 new clusters. Finally, PAcluster80 was constructed by the 56,805 clusters, which consists of 106,275 PDB and 100,912 AlphaFold DB structures.

**Structure profiles and distance profiles.** The structure profiles are histogram distributions of pairwise residue distances, which were derived from structure classes of PAcluster80 by statistical consistency analysis. The member structures of the cluster were globally aligned with the centroid by TM-align, and the distance distribution of the residue pairs of centroid structures was extracted. As shown in Supplementary Fig. S11, we divided the distance range (2–20 Å) into 36 bins with a

size of 0.5 Å, plus one bin indicating residue pair distance ≥20 Å. The number of times of falling into the bin divided by the total was taken as the probability. Note that residue pairs with gaps were not included in the total. The distance profiles were represented as the probability distribution of pairwise residue distances predicted by our in-house DeepMDisPre, an inter-residue distance predictor using a convolutional network with self-attention. The input of DeepMDisPre was the query sequence, and MSAs were generated by searching UniRef30[59] with HHblits[60]. Importantly, DeepMDisPre predicted a multi-peak distance distribution within 20 Å for flexible protein region, which provides more information for template recognition and folding pathway exploration.

**Three-track alignment.** To take full advantage of the deposited structure information to identify templates, we developed a three-track alignment (sequence alignment, residue pair alignment, and profile alignment) for two stages. In the first stage, the optimal $N_{clu}$ structural clusters are identified by the three-track alignment between the query sequence and the cluster seeds. In the second stage, the optimal templates are identified from the structures within the clusters determined in the first stage by repeating the three-track alignment.

The purpose of the three-track alignment is to find an optimal alignment between the query sequence and the template sequence by maximizing the alignScore defined by Eq. (1), as shown in Fig. 1b. The first track is to calculate the protein-specific score matrix and find the optimal sequence alignment by dynamic programming. The protein-specific score matrix is obtained from the second track by a second dynamic programming to find the optimal residue pair alignment that only considers the inter-residue distance. The residue pair alignment is performed based on the construction of the residue pair score matrix, where the values are calculated from the third track by maximizing the product of probabilities and minimizing the difference of distances.

The alignment scores of the query sequence and template are defined as follows:

$$\text{alignScore}(a, b) = \frac{S_{\text{align}}^{a,b} + S_{\text{gap}}^{a,b}}{S_{\text{tot}}^{a}} \quad (1)$$

where $a$ and $b$ are the query sequence and template, respectively; $S_{\text{align}}^{a,b}$ is the alignment score; and $S_{\text{gap}}^{a,b}$ is the penalty score for gaps, which is obtained by counting the gaps in the alignment of the query sequence and template. $S_{\text{tot}}^{a}$ is the reference score for normalization. $S_{\text{align}}^{a,b}$ and $S_{\text{tot}}^{a}$ are defined by Eq. (2) and Eq. (3), respectively:

$$S_{\text{align}}^{a,b} = \sum_{n=1}^{L} \sum_{m=1}^{L} \max\left\{ p_{n,m}^{i} \cdot q_{n,m}^{j} \cdot w_{n,m}^{\text{seq}} \cdot w_{n,m}^{\text{dist}} \cdot w_{n,m}^{\text{cut}} | i \in [1, k_a], j \in [1, k_b] \right\} \quad (2)$$

$$S_{\text{tot}}^{a} = \sum_{n=1}^{L} \sum_{m=1}^{L} p_{n,m}^{\max} \cdot w_{n,m}^{\text{seq}} \cdot w_{n,m}^{\text{cut}} \quad (3)$$

where $L$ is the length of the alignment; $p_{n,m}^{i}$ and $q_{n,m}^{j}$ are the $i$-th and $j$-th maxima of probability between the $n$-th and $m$-th residues of distance profiles and structure profiles, respectively; and $p_{n,m}^{\max}$ is the highest probability between the $n$-th and $m$-th residues of the distance profiles. $k_a$ and $k_b$ are the number of maxima of probability selected from the distance profiles and structure profiles, respectively. In the second three-track alignment stage, the structure profile only uses the single maxima because it is extracted from a single structure. $w_{n,m}^{\text{seq}}$ is the weight of sequence separation, which is designed based on the fact that long-range sequence separation is more important than short-range separation, and it is given as follows:

$$w_{n,m}^{\text{seq}} = \begin{cases} 0.5, & \text{if } |n - m| < 12 \\ 0.75, & \text{if } 12 \leq |n - m| < 24 \\ 1, & \text{otherwise} \end{cases} \quad (4)$$

where $w_{n,m}^{\text{dist}}$ is the weight of the distance difference of the residue pair between the query sequence and the template. The smaller the distance difference is, the greater the weight, which is shown as follows:

$$w_{n,m}^{\text{dist}} = \begin{cases} 1, & \text{if } |d_{n,m}^{a} - d_{n,m}^{b}| \leq 5 \text{ Å} \\ 0.5, & \text{if } 5 \text{ Å} < |d_{n,m}^{a} - d_{n,m}^{b}| \leq 8 \text{ Å} \\ 0.25, & \text{otherwise} \end{cases} \quad (5)$$

where $w_{n,m}^{\text{cut}}$ represents that only the distance information between the $n$-th and $m$-th residues within $\lambda$ Å are used, which is shown as follows:

$$w_{n,m}^{\text{cut}} = \begin{cases} 1, & \text{if } D_{n,m}^{a} < \lambda, D_{n,m}^{b} < \lambda \\ 0, & \text{otherwise} \end{cases} \quad (6)$$

$$D_{n,m}^{a} = \max\{d_{n,m}^{a}(i) | i \in [1, k_a]\}, D_{n,m}^{b} = \max\{d_{n,m}^{b}(j) | j \in [1, k_b]\} \quad (7)$$

**pDMScore prediction and ranking of templates.** According to alignScore, we selected the top $N_{\text{pDM}}$ templates to predict pDMScore. The physical and geometric features of the template structure were extracted as the input of the network. Physical features include Rosetta intra-residue energy terms, secondary structure

and amino acid properties[61]. Geometric features include multiple distances of inter-residue, orientations, voxelization and ultrafast shape recognition[62]. More detailed descriptions of the features are listed in Supplementary Table S7.

We used a convolutional network with self-attention to predict the quality score of the template. The first step is feature concatenation. The voxelization is input to a 3D convolutional layer and flattened into a 1D vector. It is concatenated with 1D features and fed into a 1D convolutional layer. The output 1D vectors are horizontally and vertically striped into 2D vectors and concatenated with 2D features. It is then turned into a 128-channel feature matrix through 2D convolution, instance normalization, and ELU activation. The second step is the self-attention operation, which contains 3 axial-attention blocks. The feature matrix is processed by an axial multi-head attention method, which alternates attention on the rows and columns of all features. The number of attention heads is 8. The encoded representation is a set of query-key-value triples in each head of the self-attention layer. The output from masked multi-head attention is further processed by a pointwise feedforward layer. The third step is the convolution operation. The convolution operation consists of 15 residual blocks, each of which is composed of three ELU activation layers, three 2D convolutional layers, and three instance normalization layers. The output layer is processed through sigmoid and softmax functions to predict the Cα distance deviations for all residual pairs.

DMScore, a global structure scoring metric proposed in our previous study[35], is used to evaluate the model scores by computing the Euclidean distance difference of the residue pairs, taking values (0, 1). DMScore not only evaluates the global topology but also considers the local structure differences, which are defined as follows:

$$\text{DMScore} = \frac{1}{L(L-1)} \sum_{i=1}^{L} \sum_{j=1}^{L} \frac{1}{1 + (d_{i,j}/d^{*})^{2}} \quad (8)$$

$$d^{*} = \log(\varepsilon + |i - j|), i \neq j \quad (9)$$

where $L$ is the length of the protein structure; $d_{i,j}$ is the predicted deviation between the $i$-th and $j$-th residues; $d^{*}$ is the normalized scale used to eliminate the inherent dependence of the score on protein size; and $\varepsilon$ is an infinitely small quantity so that $d^{*}$ is not zero.

The loss function includes the distance deviation of the residual pair and pDMScore of the residue, which are evaluated by the multivariate cross-entropy loss function and the mean square loss function, respectively. The loss function is defined as follows:

$$loss = loss_{\text{dev}} + w \cdot loss_{\text{pDM}} \quad (10)$$

where $loss_{\text{dev}}$ is the distance deviation loss and $loss_{\text{DM}}$ is the pDMScore loss of each residue.

We ranked the templates by rankScore calculated with linear weighting of alignScore and pDMScore, which is defined as follows:

$$\text{rankScore} = \text{alignScore} * \text{alignScore} + (1 - \text{alignScore}) \cdot \text{pDMScore} \quad (11)$$

**Protein folding pathway exploration.** Based on a large number of homologous templates, PAthreader explores the protein folding pathway by calculating residue frequency distributions and identifying intermediates. First, the structure of the target protein was determined with the first template. We selected the top $N_t$ templates for global structural alignment with the target protein structure using TM-align. The frequency scores of each residue were then calculated by Eq. (12), which revealed the similarities and differences between homologous templates.

$$\text{ResFscore}_i = \frac{1}{N_t} \sum_{n=1}^{N_t} \text{score}_n, i \in [1, L] \quad (12)$$

$$\text{score}_n = \begin{cases} 1, & \text{if } d_i \leq 2 \text{ Å} \\ 0.75, & \text{if } 2 \text{ Å} < d_i \leq 4 \text{ Å} \\ 0.25, & \text{if } 4 \text{ Å} < d_i \leq 5 \text{ Å} \\ 0, & \text{otherwith} \end{cases} \quad (13)$$

where $L$ is the length of the target protein; $N_t$ is the number of templates; and $d_i$ is the Euclidean distance between the $i$-th residue of the target protein and the corresponding residue of the other templates;

Second, the target protein was divided into multiple segments based on three secondary structure types (α-helix, β-sheet and loop), and loop segments ≤4 AAs were merged into neighboring helix or sheet segments. The average ResFscore for each segment was calculated by dividing the corresponding area of the segment by its length, which is defined as follows:

$$\text{ResFscore}_s^{\text{ave}} = \frac{1}{L_s} \sum_{j=1}^{L_s} \text{ResFscore}_j, s \in [1, S] \quad (14)$$

where $S$ is the number of segments; $L_s$ is the length of the $s$-th segment. The segments with a ResFscore$_s^{\text{ave}} \geq I_{\text{cut}}$ were selected as intermediates. $I_{\text{cut}}$ is a threshold parameter with an initial value of 0.4, ranging from 0 to 1. If the length of the intermediate was outside the range [0.25*$L$, 0.75*$L$], the intermediate will be re-

selected by adjusting the parameter $I_{cut}$ with a step size of 0.02 until the conditions were met, where $L$ is the length of query sequence.

**Statistics and reproducibility.** All data were carefully collected and analyzed using standard statistical methods. Comprehensive information on the statistical analyses used is included in various places, including the figures, figure legends and results, where the methods, significance and *p*-values are described.

**Reporting summary.** Further information on research design is available in the Nature Portfolio Reporting Summary linked to this article.

## Data availability

The authors declare that the data supporting the results and conclusions of this study are available within the paper and its Supplementary Information. The pDMscore model, the distance profile prediction model and the master structure database PAcluster80 are available at http://zhanglab-bioinf.com/PAthreader/.

## Code availability

The online server and package of PAthreader are made freely available at http://zhanglab-bioinf.com/PAthreader/ and GitHub (https://github.com/iobio-zjut/PAthreader).

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

## Acknowledgements

We thank Jun Liu and Zhaohong Huang for their discussions and feedback. We also thank Pengxin Zhao for helping develop the server. This work was supported by the "New Generation Artificial Intelligence" major project of Science and Technology Innovation 2030 of the Ministry of Science and Technology of the People's Republic of China [2022ZD0115100], the National Nature Science Foundation of China [62173304, 62203389], the Key Project of Zhejiang Provincial Natural Science Foundation of China [LZ20F030002], the Institute for Frontiers and Interdisciplinary Sciences of Zhejiang University of Technology [2022JCY24].

## Author contributions

G.Z. conceived and designed research. S.L. helped design research. K.Z. and Y.X. wrote algorithm and performed the experiments. G.Z., K.Z. and F.Z. developed DeepMDisPre program. G.Z., K.Z. and X.Z. analyzed data and developed the server. G.Z. and K.Z. wrote the manuscript, and all authors proofread the manuscript.

## Competing interests

The authors declare no competing interests.
