## [Peer Review File · Communications Biology]

Reviewers' comments:

Reviewer #1 (Remarks to the Author):

Undoubtedly, templates are essential for both protein structure prediction and protein folding pathways. The author hypothesized that recognition of a better template will improve the accuracy of protein structure prediction. Hence, the author introduced PAtreader, which uses a three-track alignment method to ensure maximum alignment scores, hence outputting best ranked templates. A few issues can be addressed:

- 1) The author indicated the accuracy of PAtreader is better than HHsearch and LOMETS3. Did other tools, such as SPARKS-X and MUSTER, are used for the comparison? Does PAtreader outperform those tools?
- 2) Line 168, Figure 3. The meaning of the superscript pound sign (#) should be explained in Fig. 3 Caption.
- 3) Line 378: 2021 proteins or 2021 clusters?
- 4) Line 391- PDB clusters is 34701. This is slightly different from 34700 reported in Supplementary Fig. S9.

Reviewer #2 (Remarks to the Author):

The authors present an interesting tool, PAtreader, that can be used to improve who we study and predict protein structures and how we evaluate protein folding. Included below are some suggested revisions that will help to enhance the current version of this manuscript.

Introduction:

1. The authors indicate that the understanding protein folding is important for deciphering the genetic code and will promote the exploration of pathogenic mechanisms and the development of drug designs. The authors should use this opportunity to also highlight how improvements in our understanding of protein folding can also advance the methods used to engineer proteins for other functions beyond their use as therapeutics. Can we use this knowledge to build better biomaterials for example?
2. The authors should expand more on how evolutionary relationships are contained in templates?
3. The authors should comment on how kinetic methods such as stop flow fluorescence (and others) can be used to elucidate the folding pathway of single domain proteins.
4. The authors should elaborate more on how deep learning methods could be used to leverage MD trajectories to characterize and predict protein folding pathways.
5. The authors should elaborate on how they expect the folding pathway of a single domain protein to be impacted when it is expressed as part of a fusion protein.
6. The authors should define what they mean as remote template.

Main text:

1. More detail on how the sequence of the folding intermediates is determined.
2. Why as the 80% structural similarity threshold used in Figure 1 to cluster PDB and ALphaFOld DB data?
3. Why does PAtreader perform substantially better on the hard targets (0-0.5, 0.5-0.7 subsets, Table 1) compared to all of the other subsets? How can you improve the performance of PAtreader beyond what is observed when compared to HHsearch and LOMETS3?
4. When selecting the proteins to work with did the authors select based on protein size (similar size 1dom, 2dom, 3dom in each group), loop length or any such criteria?
5. Why do the authors suspect that PAtreader performed equally well on 1dom and 2dom proteins?
6. How do the authors account for the significant drop in performance of PAtreader with the >3dom subset (see figure 2)?
7. When attempting to recognize the templates of protein complexes why was

the decision to use 10 histidine residues to link the chains of the complex together? Why not use a natural linker of equal sequence length or a 10 residue sequence containing flexible sequence repeats such as GST or GAS?

8. Is there really a large gap between HHsearch and its native alignment HHsearch#? The data presented in figure 3 suggests otherwise.

9. What is the frequency with which high scoring models were obtained without high quality templates?

10. Can the authors elaborate of the reverse trend observed in figure 4b?

11. Did the authors consider using contact maps as a way to validate/evaluate the functional significance of the data obtained via the ResFscore?

12. Can PAtreader predict the kinetic parameters of intermediate states?

13. The fact that the order of the yellow sequence regions could not be predicted accurately suggests that there is a minimum sequence length that this system needs. Can the authors identify what that sequence length is?

Reviewer #3 (Remarks to the Author):

This manuscript introduces PAtreader, a software package designed to identify remote homologs in both the PDB and the AlphaFold database. The authors show that improving the quality of templates supplied to AlphaFold improves the quality/accuracy of the models generated. Additionally, PAtreader can calculate predicted folding pathways based on the templates identified.

Overall, this is a nice manuscript and of particular interest to me was the method introduced to predict protein folding pathways.

Issues:

1. The overall quality of the English throughout the paper needs improving. E.g., page 2, lines 33-34 "There's some recent work suggests that almost all of the...".

2. Page 2, lines 42-43 - "However, it fails to learn the physics of protein to better understand the mechanisms of folding" – suggest rephrasing.

3. Page 3, lines 58-59 - the number of molecules in the PDB/AFDB are out of date by a long way.

4. Page 4, lines 89-90 - "state-of-the-art structure prediction methods" – which methods?

5. Page 9, lines 207-208 - "The other is that these test proteins might have been included in the training set of AlphaFold2, which makes it difficult to accurately test the effect of templates on AlphaFold2" – was the max_template_date parameter set when running AlphaFold2? Also, could you not avoid this problem by limiting your test set to proteins released since AlphaFold2 was trained (2022-01-19).

6. Page 9, lines 219-220 - "This is probably because PAtreader provides better templates than HHsearch for the modelling" – I've noticed the use of 'probably' in several places throughout the manuscript and I'd suggest rephrasing throughout. In this instance something like "This suggests PAtreader was able to provide better templates for the modelling than HHsearch" would read better.

7. Page 11, lines 260-263 - "Yet to be determined by biological experiments" is almost repeated by "Yet to be verified by biological experiments". This only needs to be said once.

8. Page 14, Figure 6 - The orientation of the helices in the intermediate defined by PAtreader is different for 1YYJ. This makes it hard to see if PAtreader has accurately picked out the right helices. Also in this figure, why aren't the secondary paths predicted by PAtreader shown in red?

9. Page 21, line 492 - What is the initial I_{cut} threshold value? How is the threshold adjusted?

Here are some general questions that I think ought to be answered in the paper:

1. Given that the PAtreader database has been supplemented by AlphaFold models, it is perhaps unsurprising that more useful templates are found than in other state-of-the-art methods. In Figure 3 the performance of PAtreader with and without the AlphaFold database is compared. How does the

method without the AlphaFold DB compare to HHSearch/LOMETS3?

2. The output of AlphaFold2 is said to improve when using AlphaFold database models as templates. Given that AlphaFold2 was able to accurately model the template, do you have any insight as to why AlphaFold2 requires these templates to accurately model your targets?

3. With the AlphaFold database expanding to over 200 million entries, how will this impact the method?

4. The AlphaFold database doesn't contain any information for viral proteins, does PAtreader provide any benefit over other methods for these types of targets?

5. The ESM Metagenomic Atlas was released early in November, are there plans to include these models in the PAcluster80 database?

Gui-Jun Zhang
College of Information Engineering
Zhejiang University of Technology
Hangzhou 310023, China
zgj@zjut.edu.cn
January 12 2023

Dear Reviewers,

Thank you for reviewing our manuscript entitled “Research on protein structure prediction and folding based on novel remote homologs recognition” (COMMSBIO-22-3726-T) and for proposing valuable suggestions. The comments definitely helped us improve our manuscript and served as important guidance to our research. We have carefully considered the comments and have revised the manuscript accordingly.

Thank you again for reviewing our manuscript. We hope you find that the changes are satisfactory.

Sincerely,

Kailong Zhao, Yuhao Xia, Fujin Zhang, Xiaogen Zhou, Stan Z. Li, and Guijun Zhang

Response to Reviewer 1

We very much appreciate for your comments and suggestions, which help to significantly improve the quality and description of the manuscript. In the following, we include point-by-point replies to the comments, where all changes have been highlighted in yellow in the manuscript.

Undoubtedly, templates are essential for both protein structure prediction and protein folding pathways. The author hypothesized that recognition of a better template will improve the accuracy of protein structure prediction. Hence, the author introduced PAtreader, which uses a three-track alignment method to ensure maximum alignment scores, hence outputting best ranked templates. A few issues can be addressed:

Response: We thank the Reviewer for the nice summary and positive comments on our work.

1) The author indicated the accuracy of PAtreader is better than HHsearch and LOMETS3. Did other tools, such as SPARKS-X and MUSTER, are used for the comparison? Does PAtreader outperform those tools?

Response: According to your suggestion, we have added experiments to compare with SPARKS-X, MUSTER, CEthreader and EigenTHREADER, where homologous templates with a sequence identity $\geq 30\%$ to the query were excluded. The results are summarized in **Supplementary Table. S2**. The average TM-score of PAtreader is 12.2%, 5.2%, 8.4%, 8.5%, 7.9% and 10.5% higher than that of HHsearch, LOMETS3, SPARKS-X, MUSTER, CEthreader and EigenTHREADER, respectively. The results show that the PAtreader performs better than these methods.

We have added the description and analysis to lines 133-137 of the manuscript.

of HHsearch and LOMETS3, respectively. We also compared our method with SPARKS-X, MUSTER, CEthreader and EigenTHREADER, where homologous templates with a sequence identity $\geq 30\%$ to the query were excluded. The results are summarized in **Supplementary Table. S2**. The average TM-score of PAtreader is 12.2%, 5.2%, 8.4%, 8.5%, 7.9% and 10.5% higher than that of HHsearch, LOMETS3, SPARKS-X, MUSTER, CEthreader and EigenTHREADER, respectively. The results show that the PAtreader performs better than these methods.

Table S2. TM-score of template recognition on 551 tested proteins. Tested proteins were divided into four subsets (0-0.5, 0.5-0.7, 0.7-0.9 and 0.9-1) based on TM-score of the best template of targets in PDB. Bold text highlights the best result in each category.

	(0.9, 1.0]	(0.7, 0.9]	(0.5, 0.7]	(0.0, 0.5]	All
	num (58)	num (321)	num (149)	num (23)	num (551)
PAthreader	0.899	0.787	0.568	0.424	0.725
HHsearch	0.840	0.718	0.476	0.272	0.646
LOMETS3	0.868	0.754	0.534	0.342	0.689
SPARKS-X	0.851	0.736	0.505	0.326	0.669
MUSTER	0.854	0.737	0.500	0.327	0.668
CEthreader	0.859	0.740	0.507	0.322	0.672
EigenTHREADER	0.841	0.727	0.488	0.324	0.656

2) Line 168, Figure 3. The meaning of the superscript pound sign (#) should be explained in Fig. 3 Caption.

Response: Thank you for your suggestion. We have added the description of PAthreader[#] and HHsearch[#] in the Fig. 3 Caption. PAthreader[#] and HHsearch[#] represent the results obtained by comparing the identified structure with the native structure through TM-align.

Fig. 3 Performance of PAthreader for template recognition. **a** The average TM-score for template recognition with and without AlphaFold DB at different cut-off ranges. **b, c** The proportion of the number of templates with $\geq 30\%$ and 100% sequence identity removed at different TM-score cut-off. **PAthreader[#]** and **HHsearch[#]** represent the results obtained by comparing the identified structure with the native structure through TM-align. **d** The average TM-score for different template rankings, showing the effect of pDMScore on template recognition.

3) Line 378: 2021 proteins or 2021 clusters?

Response: We apologize for the unclear description, and it should be 2,021 clusters. We have revised this in the manuscript.

templates with high sequence identity in the template library, which helps to objectively evaluate the ability of remote homologous templates recognition. **In the 2,021 clusters**, 551 nonredundant proteins were selected as the benchmark set

4) Line 391- PDB clusters is 34701. This is slightly different from 34700 reported in Supplementary Fig. S9.

Response: We thank the reviewer for pointing this out. The number of PDB clusters is 34701. We have corrected the **Supplementary Fig. S10**.

←

Figure S10. Schematic of the construction of the master structure database. PAcluster80 is constructed with 56,805 clusters consisting of 106,275 PDB structures and 100,912 AlphaFold DB structures. The structural profiles are extracted from the structural classes.↵

Response to Reviewer 2

We very much appreciate for your comments and suggestions, which help to significantly improve the quality and description of the manuscript. In the following, we include point-by-point replies to the comments, where all changes have been highlighted in yellow in the manuscript.

The authors present an interesting tool, PAtreader, that can be used to improve who we study and predict protein structures and how we evaluate protein folding. Included below are some suggested revisions that will help to enhance the current version of this manuscript.

Response: Thank you for the nice summary and positive comments on our work.

Introduction:

1) The authors indicate that the understanding protein folding is important for deciphering the genetic code and will promote the exploration of pathogenic mechanisms and the development of drug designs. The authors should use this opportunity to also highlight how improvements in our understanding of protein folding can also advance the methods used to engineer proteins for other functions beyond their use as therapeutics. Can we use this knowledge to build better biomaterials for example?

Response: Thank you for your suggestion. Protein engineering can be employed to encode protein-based materials with desired properties and functionality. As the Reviewer said, understanding protein folding is also important for protein design and protein engineering, which is fundamental to the study of engineered biomaterials. We have added the description in the introduction.

structure modelling. However, the physics of how proteins dynamically fold into their equilibrium structures is not explored in AlphaFold^{2,1,2}, while understanding protein folding is important for deciphering the genetic code and will promote the exploration of pathogenic mechanism, the development of drug design, and the design of engineered protein-based materials³⁻⁵. It is well known that templates play a critical role in the protein structure modelling⁴. Meanwhile, the

5. Connell, K.B., Miller, E.J. & Marqusee, S. The folding trajectory of RNase H is dominated by its topology and not local stability: a protein engineering study of variants that fold via two-state and three-state mechanisms. *Journal of Molecular Biology* **391**, 450-460 (2009).⁴

2) The authors should expand more on how evolutionary relationships are contained in templates?

Response: We thank you for your suggestions and we have added descriptions to lines 84-86 of the manuscript. We propose a hypothesis that the evolutionary relationship of protein families implicitly contains the folding information of individual proteins. That is, the folding pattern that occur frequently in protein structures are more evolutionarily conserved, and these conserved regions are structurally stable and preferentially formed in protein folding.

the template availability. When feeding the better template of PAtreader into AlphaFold2, the accuracy of its model was improved. Furthermore, we propose a hypothesis that the evolutionary relationship of protein families implicitly contains the folding information of individual proteins. That is, the folding pattern that occur frequently in protein structures are more evolutionarily conserved, and these conserved regions are structurally stable and preferentially formed in protein folding. Based on the above, we identified folding intermediates from homologous templates and obtained protein folding pathways

3) The authors should comment on how kinetic methods such as stop flow fluorescence (and others) can be used to elucidate the folding pathway of single domain proteins.

Response: Thank you for your suggestion. We have added the description to lines 45-49 in the introduction section. Circular dichroism spectroscopy uses the different absorption of left and right circularly polarized light by chiral molecules to analyze protein interactions and determine protein folding paths. Fluorescence spectroscopy is based on the fluorescence emission characteristics of aromatic amino acids in proteins to monitor the folding-unfolding pattern and protein denaturation.

folding²¹. Experimentalists usually use spectrometry methods such as circular dichroism chromatography, fluorescence spectroscopy, etc. to study protein folding pathways. Circular dichroism spectroscopy uses the different absorption of left and right circularly polarized light by chiral molecules to analyze protein interactions and determine protein folding paths²². Fluorescence spectroscopy is based on the fluorescence emission characteristics of aromatic amino acids in proteins to monitor the folding-unfolding pattern and protein denaturation²³. These methods can follow kinetic folding but provide very

4) The authors should elaborate more on how deep learning methods could be used to leverage MD trajectories to characterize and predict protein folding pathways.

Response: We have added the description to lines 50-54 in the introduction section. Kresten Lindorff-Larsen et al. studied the folding process of proteins through equilibrium MD simulations by improving the CHARMM force field to make it easier to transfer between different protein classes [1]. Charlotte M Deane et al. use DMPfold to predict the distribution of distances for each residue pair to determine rigid and flexible behavior of proteins [2].

limited information to define the structure of folding intermediates²⁴. In some related works, attempts have been made to avoid these difficulties by simulating the folding process. Kresten Lindorff-Larsen et al. studied the folding process of proteins through equilibrium MD simulations by improving the CHARMM force field to make it easier to transfer between different protein classes²¹. Charlotte M Deane et al. use DMPfold to predict the distribution of distances for each residue pair to determine rigid and flexible behavior of proteins²⁵. However, inferring the folding path from the mass of simulated data is

[1] Lindorff-Larsen, K., Piana, S., Dror, R.O. & Shaw, D.E. How Fast-Folding Proteins Fold. *Science* **334**, 517-520 (2011).

[2] Schwarz, D. et al. Co-evolutionary distance predictions contain flexibility information. *Bioinformatics* **38**, 65-72 (2022).

5) The authors should elaborate on how they expect the folding pathway of a single domain protein to be impacted when it is expressed as part of a fusion protein.

Response: Our method is a computational method for protein folding pathway prediction based on remote homologues. We propose a hypothesis that the evolutionary relationship of protein families implicitly contains the folding information of individual proteins. That is, the folding pattern that occur frequently in protein structures are more evolutionarily conserved, and these conserved regions are structurally stable and preferentially formed in protein folding. Based on the above, we obtained the correct folding pathway on seven widely studied proteins and verified our hypothesis.

Topology of protein conformation, or the spatial arrangement of structural units and the chain connectivity among them, is a key determinant of the folding mechanisms of proteins. When a protein contains multiple regions of cooperative structure formation (i.e., foldons or domains), the folding path of the protein should be determined by the interactions between structural regions and the strength of cooperation within a single region. Magnus O Lindberg et al. have suggested that the protein folding pathway is determined by the composition of nucleation-competent submotifs and how they are coupled [3]. If the protein structure contains only one foldon, the pathway is relatively robust and the transition state plasticity small. When there are multiple, overlapping foldons, the folding progression can be broad and partitioned over multiple pathways. The folding pathway of fusion proteins is an interesting and important work that we will consider next.

[3] Hartl, F.U. and Hayer-Hartl, M. Converging concepts of protein folding in vitro and in vivo. *Nature structural & molecular biology*, 16, 574-581(2009).

6) The authors should define what they mean as remote template.

Response: Thank you for your suggestion. In this study, PDB structures with a sequence identity < 30% were defined as remote homologous templates[4-5]. We have added the description to lines 39-40.

solve most single-domain proteins¹⁷. Therefore, developing a new method to recognize high-quality remote homologous templates (PDB structures with a sequence identity < 30%)^{18,19} is extremely important for protein structure prediction.⁴

[4] Steinegger, M. et al. HH-suite3 for fast remote homology detection and deep protein annotation. *BMC Bioinformatics* **20** (2019).

[5] Zheng, W. et al. Detecting distant-homology protein structures by aligning deep neural-network based contact maps. *PLoS Comput. Biol.* **15** (2019).

Main text:

1) More detail on how the sequence of the folding intermediates is determined.

Response: We explore protein folding pathways based on homologous templates and secondary structures, consisting of two consecutive steps. First, the global structural alignment of the identified homologous templates with the first template was performed by TM-align. The residue frequency scores (ResFscore) were then calculated at different distance deviation thresholds, where a higher value indicates a higher frequency of residue alignment at corresponding positions of template structures. Second, the target was divided into multiple segments based on the secondary structure, and the average ResFscore for each segment was calculated by dividing the corresponding area of the segment by its length. Folding intermediates were finally determined by adaptively selecting fragments with an average ResFscore greater than a given threshold.

We have added the description to lines 289-292 and 524-526 of the manuscript.

steps. First, the global structural alignment of the identified homologous templates with the first template was performed by TM-align. The residue frequency scores (ResFscore) were then calculated at different distance deviation thresholds, where a higher value indicates a higher frequency of residue alignment at corresponding positions of template structures. Second, the target was divided into multiple segments based on the secondary structure, and the average ResFscore for each segment was calculated by dividing the corresponding area of the segment by its length. Folding intermediates were finally determined by adaptively selecting fragments with an average ResFscore greater than a given threshold (see details in Methods section). We

where S is the number of segments; L_s is the length of the s -th segment. The segments with a $\text{ResFscore}_s^{\text{ave}} \geq I_{\text{cut}}$ were selected as intermediates. I_{cut} is a threshold parameter with an initial value of 0.4, ranging from 0 to 1. If the length of the intermediate was outside the range $[0.25*L, 0.75*L]$, the intermediate will be re-selected by adjusting the parameter I_{cut} with a step size of 0.02 until the conditions were met, where L is the length of query sequence. ⁴

2) Why as the 80% structural similarity threshold used in Figure 1 to cluster PDB and AlphaFold DB data?

Response: This is because two proteins with a TM-score ≥ 80 have very similar topologies. Zhang et al. have reported the posterior probability of proteins with a given TM-score being in the same Fold or in different Fold families. As shown in **Figure R1**, when the TM-score of two proteins is < 0.4 , there are almost no protein pairs belonging to the same SCOP Fold family. When TM-score ≥ 0.8 , the probability of two proteins in the same SCOP Fold is almost 100%^[6].

Figure R1. The posterior probability of proteins with a given TM-score being in the same Fold. (squares, triangles and stars points) or different Fold family (circle points). (Image is from [1]).

[6] Xu, J & Zhang, Y. How significant is a protein structure similarity with TM-score = 0.5?. *Bioinformatics* 7, 889-95 (2010).

We have added the description to lines 407-408 of the manuscript.

106,275 proteins using TM-align and classified them into 34,701 structural classes based on an 80% structural similarity threshold (protein pairs with TM-score ≥ 0.8 have a great probability of being the same SCOP fold, and they have very similar topologies)⁵⁸. In this process, we used a greedy incremental clustering approach similar to CD-HIT, which avoids

58. Xu, J. & Zhang, Y. How significant is a protein structure similarity with TM-score= 0.5? *Bioinformatics* 26, 889-895 (2010).⁴

3) Why does PAtreader perform substantially better on the hard targets (0-0.5, 0.5-0.7 subsets, Table 1) compared to all of the other subsets? How can you improve the performance of PAtreader beyond what is observed when compared to HHsearch and LOMETS3?

Response: Thank you for your suggestion.

- 1) The 0-0.5, 0.5-0.7 subsets perform better than other subsets because AlphaFold DB complements the PDB library by providing novel fold structures and motifs that are not in the PDB. When AlphaFold DB is used, the average TM-score of PAtreader is 0.1%, 1%, 8.2% and 23.4% higher than those without using AlphaFold DB for the targets of 0.9-1, 0.7-0.9, 0.5-0.7 and 0-0.5, respectively, where the performance of PAtreader is significantly improved for the hard targets (0-0.5 and 0.5-0.7).
- 2) HHsearch aligns the query HMM with each of the target HMMs using the Viterbi dynamic programming algorithm to find the alignment with the maximum score. LOMETS3 is a meta-server approach that integrates multiple deep learning-based threading methods and state-of-the-art profile-based programs. PAtreader performs template recognition based on the three-track alignment between predicted distance profiles and structure profiles. As shown in Figure 3, there are three reasons for the better performance of PAtreader over HHsearch and LOMETS3: a) AlphaFold DB helps to improve the accuracy of template recognition by expanding the family coverage of model organism proteomes. b) The three-track alignment

algorithm provides accurate sequence-template alignment. c) The predicted pDMScore helps to select physically plausible templates.

4) When selecting the proteins to work with did the authors select based on protein size (similar size 1dom, 2dom, 3dom in each group), loop length or any such criteria?

Response: This is a good suggestion. In this study, our test set was constructed based on protein size, resolution, and redundancy. The benchmark set was constructed from SCOPe 2.07, which was divided into 11,198 clusters by CD-HIT with a 30% sequence identity cut-off. We selected 2,021 clusters with only one member as candidate sets because they have few templates with high sequence identity in the template library, which helps to objectively evaluate the ability of remote homologous templates recognition. In the 2,021 clusters, 551 nonredundant proteins were selected as the benchmark set based on sequence lengths ranging from 120 to 700 AAs and resolutions less than or equal to 2.0 Å. In future studies, we will consider loop length as one of the criteria for constructing test sets.

5) Why do the authors suspect that PAtreader performed equally well on 1dom and 2dom proteins?

6) How do the authors account for the significant drop in performance of PAtreader with the >3dom subset (see figure 2)?

Response: Both of the questions mentioned by the reviewers are related to the number of templates in the database. **Figure R2b** shows the distribution of multi-domain proteins in the PDB [7]. Statistics show that single-domain proteins and 2-domain proteins account for 64% and 27% of PDB, respectively, whereas ≥ 3 -domain proteins only account for 9%. These indicate that single-domain and 2-domain proteins have more template structures than ≥ 3 -domain proteins. This resulted in the PAtreader performing equally well on single-domain and 2-domain proteins, while significantly drop on ≥ 3 -domain proteins (Image is from [7]).

Figure R2. Distribution of multi-domain proteins in the PDB.

[7] Zhou, X. et al. Progressive assembly of multi-domain protein structures from cryo-EM density

7) When attempting to recognize the templates of protein complexes why was the decision to use 10 histidine residues to link the chains of the complex together? Why not use a natural linker of equal sequence length or a 10 residue sequence containing flexible sequence repeats such as GST or GAS?

Response: We thank the reviewer for pointing out this issue. Linking complexes with histidine is an alternative method that we know from academic reports. We have modified the description and retested the complex for template recognition. We added a 21 residue repeated Glycine-Glycine-Serine linker between each chain according to AlphaFold-Multimer [8], and then ran it as a single chain through PAtreader for template recognition. The results are shown in **Supplementary Figure S1**, which has the same precision as template identified with 10 histidine linker.

←

Figure S1. Monomeric templates (blue) detected by PAtreader for complexes 1AB9 and 1GRN.↵

Moreover, we attempted to recognize the templates of protein complexes by adding 21 residue repeated Glycine-Glycine-Serine to link the chains of the complexes together⁴⁰. **Supplementary Fig. S1** shows that templates with TM-scores = 0.97

[8] Evans R, O'Neill M, Pritzel A, et al. Protein complex prediction with AlphaFold-Multimer. *BioRxiv*, 2022: 2021.10. 04.463034.

8) Is there really a large gap between HHsearch and its native alignment HHsearch#? The data presented in figure 3 suggests otherwise.

Response: The gap between the HHsearch and its native alignment HHsearch# does exist. In order to visually compare the HHsearch/PAtreader with their native alignment, we recalculated the average TM-score of templates on 551 test proteins. The results are shown in **Figure R3**. When removing homologous templates with 100% sequence identity, the average TM-score of PAtreader and HHsearch were 0.780 and 0.686, which were 1% and 3.8% lower than for PAtreader# and HHsearch#, respectively. When removing homologous templates with 30%

sequence identity, PAtreader and HHsearch had average TM-scores of 0.725 and 0.646, which were 1.2% and 4.3% lower than PAtreader[#] and HHsearch[#], respectively. These results suggest that PAtreader is closer to its native alignment than HHsearch.

Figure R3. The average TM-score of first template with $\geq 30\%$ and 100% sequence identity removed. PAtreader[#] and HHsearch[#] represent the results obtained by comparing the identified structure with the native structure through TM-align.

9) What is the frequency with which high scoring models were obtained without high quality templates?

Response: AlphaFold2 is a very sophisticated fold recognition algorithm that exploits the completeness of the library of single domain PDB structures. We analyzed the relationship between the model accuracy and the first template used for AlphaFold2 modelling on 551 test proteins, where all templates with sequence identity $\geq 30\%$ were removed. On the 274 targets with higher quality templates (TM-score ≥ 0.7), models with TM-score ≥ 0.9 are generated for 88% of targets by AlphaFold2. On 277 targets with relatively poor templates (TM-score < 0.7), the number of models with TM-score ≥ 0.9 predicted by AlphaFold2 decreased to 49.8%. These results show that the quality of model mainly depends on the template availability.

We also analyzed the relationship between the model accuracy and the first template used for PAtreader modelling on 186 CAMEO targets. On the 62 targets with poor quality templates (TM-score < 0.7), models with TM-score ≥ 0.9 are generated for 38.7% of targets by PAtreader. On 124 targets with higher templates (TM-score ≥ 0.7), the number of models with TM-score ≥ 0.9 predicted by PAtreader increased to 83.1%.

We have added the description and analysis to lines 241-244 of the manuscript.

almost twice as high as HHsearch (17.2%). We analyzed the relationship between the model accuracy and the first template used for PAtreader modelling on CAMEO targets. On the 62 targets with poor quality templates (TM-score < 0.7), models with TM-score ≥ 0.9 are generated for 38.7% of targets by PAtreader. On 124 targets with higher templates (TM-score ≥ 0.7), the number of models with TM-score ≥ 0.9 predicted by PAtreader increased to 83.1%. Interestingly, we find that

10) Can the authors elaborate of the reverse trend observed in figure 4b?

Response: **Figure 4b** shows the distribution of the TM-score of templates identified by PAtreader and HHsearch on 186 cameo proteins. The bin of [0.9, 1] indicates the proportion of templates whose TM-score is greater than or equal to 0.9 and less than or equal to 1, and so on. The sum of the proportions of all bin is 1. Therefore, the proportion of the high score bins of PAtreader will increase with the decrease of the proportion of the low score bins, and the proportion of the high score bins of HHsearch will decrease with the increase of the proportion of the low score bins.

Fig. 4 b The distribution of the TM-score of templates identified by PAtreader and HHsearch.

11) Did the authors consider using contact maps as a way to validate/evaluate the functional significance of the data obtained via the ResFscore?

Response: We thank the reviewer for these insightful questions. We have added experiments to use the contact maps as a way to validate the functional significance of data obtained via the ResFscore. The results are presented in **Figure R4**. The left side of the figure R4 are the folding path determined by biological experiments. The middle of the figure R4 are the contact map corresponding to the structure on the left (blue is the contact map of the intermediate). The right side of the figure R4 are the contact map obtained according to $Con_frequency_{i,j}$ from top N_t templates. $Con_frequency_{i,j}$ is defined as follows:

$$Con_frequency_{i,j} = \frac{1}{N_t} \sum_{n=1}^{N_t} C_{i,j}, \quad i, j \in [1, L] \quad (1)$$

$$C_{i,j} = \begin{cases} 1, & \text{if } d_{i,j} \leq 8\text{\AA} \\ 0, & \text{if } d_{i,j} > 8\text{\AA} \end{cases} \quad (2)$$

where L is the length of the protein structure; $N_t = 500$; $d_{i,j}$ is the Euclidean distance between the i -th and j -th residues.

For the targets 1BE9, 1NTI, 1DKT and 1MBC, the contact maps obtained from the top N_t

templates are almost consistent with intermediates identified by biological experiments. However, for the targets 2LZM, 1YYJ and 1I5T, the contact maps obtained from the top N_t templates are more similar to the contact map of the whole structure of the target. This is because the local differences between the template structures and the target structure cannot be represented on the rough contact map. These results suggest that contact maps can help validate data obtained via the ResFscore to some extent, while also showing that better results can be obtained using ResFscore.

Figure R4. The left side of the figure are the folding path determined by biological experiments. The middle of the figure are the contact map corresponding to the structure on the left (blue is the contact map of the intermediate). The right side of the figure are the contact map obtained according to $Con_frequency_{i,j}$.

12) Can Pathreader predict the kinetic parameters of intermediate states?

Response: Thank you for your suggestion. Pathreader is a computational method for protein folding pathway prediction based on remote homologues and does not predict the kinetic parameters of intermediate states. However, it is an effective tool to complement or validate the protein folding pathway determined by biological experiments. In future Pathreader development, we will consider the prediction of the kinetic parameters of intermediate states.

13) The fact that the order of the yellow sequence regions could not be predicted accurately suggests that there is a minimum sequence length that this system needs. Can the authors identify what that sequence length is?

Response: We apologize for the confusion. Yawen Bai et al reported that the folding order of the yellow region of protein 1YYJ could not be defined unambiguously by biological experiments due to the lack of sufficient probes. Therefore, we cannot get the correct folding path of the yellow region from the report. The PAtreader system is theoretically not limited by the length of the protein sequence. But due to the minimum template limitations in the library and our limited computational power, the PAtreader system recommends a sequence length of 50-1000.

Response to Reviewer 3

We very much appreciate for your comments and suggestions, which help to significantly improve the quality and description of the manuscript. In the following, we include point-by-point replies to the comments, where all changes have been highlighted in yellow in the manuscript.

This manuscript introduces PAtreader, a software package designed to identify remote homologs in both the PDB and the AlphaFold database. The authors show that improving the quality of templates supplied to AlphaFold improves the quality/accuracy of the models generated. Additionally, PAtreader can calculate predicted folding pathways based on the templates identified.

Overall, this is a nice manuscript and of particular interest to me was the method introduced to predict protein folding pathways.

Response: We are grateful for reviewer's nice summary and positive comments.

Issues:

1) The overall quality of the English throughout the paper needs improving. E.g., page 2, lines 33-34 "There's some recent work suggests that almost all of the...".

Response: Thanks for your suggestion. We have revised the description in lines 33-34 of the manuscript. In addition, we have thoroughly checked the full manuscript and asked our colleague to help check the English.

RoseTTAFold¹² and RGN²¹³, template information is used explicitly or implicitly for deep learning models. Some recent work have shown that almost all of the popular protein structure prediction methods strongly depend on the quality of the

2) Page 2, lines 42-43 - "However, it fails to learn the physics of protein to better understand the mechanisms of folding" – suggest rephrasing.

Response: We have revised the description in lines 42-43 of the manuscript.

protein structure prediction using statistical knowledge of the crystal structure. However, it is not clear whether AlphaFold2 can learn the physics of how proteins dynamically fold into equilibrium structure^{1,20}. Protein folding is a very fast process

3) Page 3, lines 58-59 - the number of molecules in the PDB/AFDB are out of date by a long way.

Response: Thank you for point out this. We have updated PAcluster80 with the latest PDB (as of December 2022) by clustering a threshold of 80% structural similarity. The number of clusters of PAcluster80 increased from 56,805 to 62987. The latest PAcluster80 can be downloaded at <http://zhanglab-bioinf.com/PAtreader>. The latest AlphaFold DB release contains over 200 million entries. In future PAtreader development, we will consider updating these structures into the PAcluster80 database to improve PAtreader performance.

4) Page 4, lines 89-90 - “state-of-the-art structure prediction methods” – which methods?

Response: We apologize for the unclear description. The state-of-the-art structure prediction method is AlphaFold2. We have revised the description in the manuscript.

physical and geometric features of the alignment structures are fed into a trained deep learning model to predict the pDMScore and rank the templates. Finally, the identified templates are integrated into AlphaFold2 for the structure modelling, and the protein folding pathway

5) Page 9, lines 207-208 - “The other is that these test proteins might have been included in the training set of AlphaFold2, which makes it difficult to accurately test the effect of templates on AlphaFold2” – was the max_template_date parameter set when running AlphaFold2? Also, could you not avoid this problem by limiting your test set to proteins released since AlphaFold2 was trained (2022-01-19).

Response: The max_template_date parameter is set to 2022-03-01 when we run AlphaFold2. According to your suggestion, we used the CAMEO test set (2022/04/01-2022/06/18) to re-analyze the relationship between the template and the accuracy of the AlphaFold2 model. The results are presented in **Supplementary Fig. S4b**. The CAMEO test set has the same trend as the 551 test proteins in terms of the relationship between the template and the accuracy of the AlphaFold2 model. AlphaFold2 produced more and more low-scoring models as template quality decreased.

Figure S4. a, b The relationship between model accuracy and template quality of AlphaFold2 on 551 test proteins and 188 cameo proteins, respectively. In the modeling of AlphaFold2, templates with $\geq 30\%$ sequence identity were removed. We selected the first template of HHsearch to compare with AlphaFold2's first model. The first template is obtained by ranking Sum_probs values of HHsearch templates. Sum_probs is the sum over the posterior probabilities of all aligned pairs of match states, which usually correspond to the template with the highest accuracy².

We have added the description and analysis to lines 222-226 of the manuscript.

AlphaFold2, which makes it difficult to accurately test the effect of templates on AlphaFold2⁴. We also used the CAMEO test set (2022/04/01-2022/06/18) to analyze the relationship between the template and the accuracy of the AlphaFold2 model. The results are presented in **Supplementary Fig. S4b**. The CAMEO test set has the same trend as the 551 test proteins in terms of the relationship between the template and the accuracy of the AlphaFold2 model. AlphaFold2 produced more and more low-scoring models as template quality decreased.↵

6) Page 9, lines 219-220 - “This is probably because PAtreader provides better templates that HHsearch for the modelling” – I’ve noticed the use of ‘probably’ in several places throughout the manuscript and I’d suggest rephrasing throughout. In this instance something like “This suggests PAtreader was able to provide better templates for the modelling than HHsearch” would read better.

Response: We apologize for this. We have revised the description in lines 195 and 237-238 of the manuscript.

This is because pDMScore uses a deep neural network that combines physical and geometric features of structures, reducing the noise from AlphaFold DB and the predicted distance profiles and effectively complementing to the alignScore. Therefore, model than other methods on most of the cases and achieves higher TM-score compared to other methods on average. This suggests PAtreader was able to provide better templates for the modelling than HHsearch. Fig. 4b presents the template

7) Page 11, lines 260-263 – “Yet to be determined by biological experiments” is almost repeated by “Yet to be verified by biological experiments”. This only needs to be said once.

Response: Thanks for pointing out this issue. We have revised the description in the manuscript. explored protein folding pathways on 7 widely studied cases and 30 human proteins. The results show that the 7 proteins are almost consistent with biological experiments, and the other 30 human proteins have yet to be verified by biological experiments.↵

8) Page 14, Figure 6 - The orientation of the helices in the intermediate defined by PAtreader is different for 1YYJ. This makes it hard to see if PAtreader has accurately picked out the right helices. Also in this figure, why aren’t the secondary paths predicted by PAtreaded shown in red?

Response: We apologize for the unclear description, which causes by the different orientations of the model shown in PyMOL. We have modified the orientation of 1YYJ and shown the secondary paths predicted by PAtreader in red in Figure 6d.

Fig. 6 Results of protein folding pathways. **a, b** Folding pathway determined by biological experiments. The folding order is blue and then red. **c** The residue frequency distribution identified by PAtreader. **d** Folding pathway determined by PAtreader. **e** Template structures with folding intermediates (blue) that are similar to those of the target protein (grey). TM-score_{local} is the similarity between the local structure (blue) of the template and the target protein.⁶⁴

9) Page 21, line 492 - What is the initial I_{cut} threshold value? How is the threshold adjusted?

Response: We apologize for the unclear description. All parameters of PAtreader are listed in **Supplementary Table S6**. The initial value of I_{cut} is 0.4 and ranges from 0 to 1. When the intermediate length outside the range of $0.75 \cdot L$, the intermediate is re-selected by increasing the I_{cut} (step size is 0.02), and when the intermediate length falls below the range of $0.25 \cdot L$, the intermediate is re-selected by decreasing the I_{cut} .

We have revised the description to lines 524-526 of the manuscript.

selected as intermediates. I_{cut} is a threshold parameter with an initial value of 0.4, ranging from 0 to 1. If the length of the intermediate was outside the range $[0.25*L, 0.75*L]$, the intermediate will be re-selected by adjusting the parameter I_{cut} with a step size of 0.02 until the conditions were met, where L is the length of query sequence. ⁴

Here are some general questions that I think ought to be answered in the paper:

1) Given that the PAtreader database has been supplemented by AlphaFold models, it is perhaps unsurprising that more useful templates are found than in other state-of-the-art methods. In Figure 3 the performance of PAtreader with and without the AlphaFold database is compared. How does the method without the AlphaFold DB compare to HHsearch/LOMETS3?

Response: Thank you for the good suggestion. We have added experiments in **Supplementary Table S1** to compare HHsearch /LOMETS3 and PAtreader without AlphaFold DB. The average TM-score of the first template identified by PAtreader is 0.702, which is 8.6% and 1.9% higher than that of HHsearch and LOMETS3, respectively.

We have added the description and analysis to lines 130-133 of the manuscript.

PAtreader is significantly better than that of HHsearch and LOMETS3 for the hard targets (0-0.5 and 0.5-0.7). **We presented experiments in Supplementary Table S1 to compare HHsearch/LOMETS3 and PAtreader without AlphaFold DB. The average TM-score of the first template identified by PAtreader is 0.702, which is 8.6% and 1.9% higher than that of HHsearch and LOMETS3, respectively. We also compared our method with SPARKS-X, MUSTER, CETHREADER and**

Table S1. Summary of the results of the ablation experiments. The results are obtained by computing the TM-score of the first template.⁴

	(0.9, 1.0] ⁴	(0.7, 0.9] ⁴	(0.5, 0.7] ⁴	(0.0, 0.5] ⁴	All ⁴
Contribution of AlphaFold DB ⁴					
PAtreader ⁴	0.899 ⁴	0.787 ⁴	0.568 ⁴	0.424 ⁴	0.725 ⁴
PAtreader (w/o AlphaFold DB) ⁴	0.898 ⁴	0.775 ⁴	0.521 ⁴	0.324 ⁴	0.702 ⁴
HHsearch ⁴	0.840 ⁴	0.718 ⁴	0.476 ⁴	0.272 ⁴	0.646 ⁴
LOMETS3 ⁴	0.868 ⁴	0.754 ⁴	0.534 ⁴	0.342 ⁴	0.689 ⁴
Contribution of pDMScore ⁴					
PAtreader ⁴	0.899 ⁴	0.787 ⁴	0.568 ⁴	0.424 ⁴	0.725 ⁴
PAtreader (alignScore) ⁴	0.896 ⁴	0.781 ⁴	0.562 ⁴	0.408 ⁴	0.718 ⁴
PAtreader (pDMScore) ⁴	0.878 ⁴	0.774 ⁴	0.557 ⁴	0.422 ⁴	0.712 ⁴

2) The output of AlphaFold2 is said to improve when using AlphaFold database models as templates. Given that AlphaFold2 was able to accurately model the template, do you have any insight as to why AlphaFold2 requires these templates to accurately model your targets?

Response: The number of structural patterns or family types of proteins is limited. We have found that many proteins with low sequence identity correspond to the same structural pattern in

PAcluster80. Therefore, templates will be essential for AlphaFold2 modelling when a protein does not have enough MSAs to infer atomic coordinates and its corresponding structural pattern happens to exist. These structural patterns are contributed by PDB and AlphaFold DB.

We have revised the description to lines 247-249 of the manuscript.

The number of structural patterns or family types of proteins is limited. Many proteins with low sequence identity correspond to the same structural pattern in PAcluster80. Therefore, templates are essential for AlphaFold2 modelling when a protein does not have enough MSAs to infer atomic coordinates and its corresponding structural pattern happens to exist. Fig.

3) With the AlphaFold database expanding to over 200 million entries, how will this impact the method?

Response: Thanks for your suggestion. The extension of AlphaFold DB will improve the accuracy of template recognition for Pathreader, because AlphaFold DB greatly expands the coverage of the sequence space and provides novel folding architectures and motifs not found in PDB. Meanwhile, the extension of AlphaFold DB provides more remote templates for folding pathway prediction, which will be beneficial to the exploitation of evolutionary relationships of protein families.

to optimize the alignment, which allows templates that satisfy all possible distance constraints to be recognized. The exploration of folding pathways based on remote templates provides new ideas and insights for the study of protein folding mechanisms, which will be advanced by the accumulation of structures deposited in PDB and the rapid expansion of AlphaFold DB.

4) The AlphaFold database doesn't contain any information for viral proteins, does Pathreader provide any benefit over other methods for these types of targets?

Response: We have added experiments to compare Pathreader with other methods on 17 proteins of SARS-CoV-2 virus, as shown in **Supplementary Table S4**. The average TM-score of Pathreader templates is 0.725, which is 11.3% higher than that of HHsearch. In structural modeling, Pathreader models achieved a slightly higher average TM-score (0.825) than the AlphaFold2 models (0.812). In **Supplementary Fig. S5**, we show a comparison of structural model built by Pathreader and AlphaFold2, where the Pathreader model has the TM-score of 0.987 which is better than that of the AlphaFold2 model (TM-score=0.825). These results demonstrate the effectiveness and robustness of Pathreader.

Table S4. Summary of the results of 17 proteins of SARS-CoV-2 virus.

	Structure modelling		Template recognition	
	PAthreader	AlphaFold2	PAthreader	HHsearch
Helicase	0.968	0.969	0.945	0.945
Proteinase 3CL-PRO	0.981	0.964	0.993	0.985
nucleocapsid protein	0.962	0.917	0.961	0.749
Non-structural protein 7	0.953	0.861	0.952	0.573
Uridylate-specific endoribonuclease	0.989	0.967	0.975	0.081
ORF7a	0.943	0.943	0.924	0.924
Non-structural protein 9	0.877	0.873	0.841	0.789
Papain-like proteinase	0.995	0.991	0.995	0.954
Non-structural protein 10	0.171	0.176	0.191	0.170
Papain-like proteinase	0.987	0.825	0.985	0.861
ORF3a	0.812	0.800	0.355	0.186
nucleocapsid protein	0.974	0.955	0.836	0.836
Non-structural protein 8	0.580	0.390	0.419	0.575
ORF8	0.410	0.749	0.270	0.208
Envelope Protein Transmembrane Domain	0.607	0.602	0.599	0.610
NSP1	0.947	0.947	0.356	0.896
Papain-like proteinase	0.874	0.870	0.731	0.732
Average TM-score	0.825	0.812	0.725	0.651

Figure S5. An illustrative example from Papain-like proteinase of SARS-CoV-2 virus, showing the structure superposition of the PAthreader model (blue) and AlphaFold2 model (red) with the native structure (yellow).

We have added experiments and descriptions of viral proteins to lines 248-254 of the manuscript.

Since 2019, considerable efforts have been made to determine the structure of proteins in SARS-CoV-2, a novel coronavirus responsible for the COVID-19 pandemic. We used PAtreader to recognize templates and model structures for 17 proteins of SARS-CoV-2 virus, and the results are shown in **Supplementary Table S4**. The average TM-score of PAtreader templates is 0.725, which is 11.3% higher than that of HHsearch. In structural modeling, PAtreader models achieved a slightly higher average TM-score (0.825) than the AlphaFold2 models (0.812). In **Supplementary Fig. S5**, we show a comparison of structural model built by PAtreader and AlphaFold2, where the PAtreader model has the TM-score of 0.987 which is better than that of the AlphaFold2 model (TM-score=0.825).⁴

5) The ESM Metagenomic Atlas was released early in November, are there plans to include these models in the PAcluster80 database?

Response: Thanks for your suggestion. The ESM metagenomic Atlas, with more than 617 million structures, was constructed by ESMFold, which uses a large language models to rapidly infer structures from primary sequence. In future PAtreader development, we will conduct large-scale analysis of the database and consider updating these structures into the PAcluster80 database to improve PAtreader performance.

REVIEWERS' COMMENTS:

Reviewer #2 (Remarks to the Author):

The authors did a good job implementing the feedback received and as a result were able to improve the overall quality and readability of the submitted manuscript.

Reviewer #3 (Remarks to the Author):

The authors have submitted a much improved manuscript and I'm content that the changes made have addressed the issues I raised in my previous review.